# Protection of metal interfaces against hydrogen-assisted cracking

Guillaume Hachet [1,2] ✉, Shaolou Wei [1], Ali Tehranchi [1,3], Xizhen Dong[1], Jérémy Lestang[2], Aochen Zhang[4], Binhan Sun [4,5] ✉, Stefan Zaefferer[1], Baptiste Gault[1,2], Dirk Ponge[1] ✉ & Dierk Raabe [1]

Enabling a hydrogen economy requires the development of materials resistant to hydrogen embrittlement (HE). More than 100 years of research have led to several mechanisms and models describing how hydrogen interacts with lattice defects and leads to mechanical property degradation. However, solutions to protect materials from hydrogen are still scarce. Here, we investigate the role of interstitial solutes in protecting critical crystalline defects sensitive to hydrogen. Ab initio calculations show that boron and carbon in solid solutions at grain boundaries can efficiently prevent hydrogen segregation. We then realized this interface protection concept on martensitic steel, a material strongly prone to HE, by doping the most sensitive interfaces with different concentrations of boron and carbon. These segregations, in addition to stress relaxations, critically reduce the hydrogen ingress by half, leading to an unprecedented resistance against HE. This tailored interstitial segregation strategy can be extended to other metallic materials susceptible to hydrogen-induced interfacial failure.

Hydrogen, either used as an energy carrier in transportation, power generation, or industrial processes, has the potential to be a key element for the reduction of carbon dioxide emissions through, for instance, the direct reduction of iron ores or the conversion of $CO_2$ into fuels[1–4]. However, metals and alloys used for hydrogen transport and storage are subject to hydrogen embrittlement, a drop in ductility, toughness, and fatigue resistance that can lead to sudden catastrophic failure of parts. Mechanisms have been proposed to explain and predict hydrogen embrittlement, including hydrogen-enhanced decohesion, hydrogen-enhanced localized plasticity, absorption-induced dislocation-emission, hydrogen-enhanced strain-induced vacancies, hydrogen-induced phase transformation, and so on[5–7]. Because of its small size, hydrogen ingress into materials is unavoidable, and overcoming HE is becoming an important scientific challenge in the field of materials science to enable a safe hydrogen economy. Recent works

have developed different solutions to mitigate HE by, for instance, refining grain size[8], forming secondary phases for hydrogen trapping[9], or developing microstructure to stop crack initiation and growth induced by hydrogen[10]. However, distinct solutions to directly protect specific microstructural features that are particularly susceptible to hydrogen remain largely unexplored.

With nearly 1.9 billion tons produced each year[11], steel is undeniably a backbone material of the world's infrastructure, with versatile properties, resulting from nearly unlimited microstructure manipulation possibilities via adjustments to the chemical composition and the cornucopian of thermomechanical processing routes that can be used[12]. Steel variants suffer from different degrees of HE. Figure 1a plots the ultimate tensile strength of a range of steels as a function of the hydrogen-induced ductility loss obtained from the difference in total elongation with and without hydrogen divided by the total

[1]Max Planck Institute for Sustainable Materials, Düsseldorf, Germany. [2]Univ Rouen Normandie, INSA Rouen, Normandie, CNRS, Normandie Univ GPM UMR 6634, Rouen, France. [3]Federal Institute for Materials Research and Testing, Berlin, Germany. [4]Key Laboratory of Pressure Systems and Safety, Ministry of Education, School of Mechanical and Power Engineering, East China University of Science and Technology, Shanghai, China. [5]State Key Laboratory of Chemical Safety, East China University of Science and Technology, East China University of Science and Technology, Shanghai, China. ✉e-mail: guillaume.hachet@cnrs.fr; binhan.sun@ecust.edu.cn; d.ponge@mpie.de

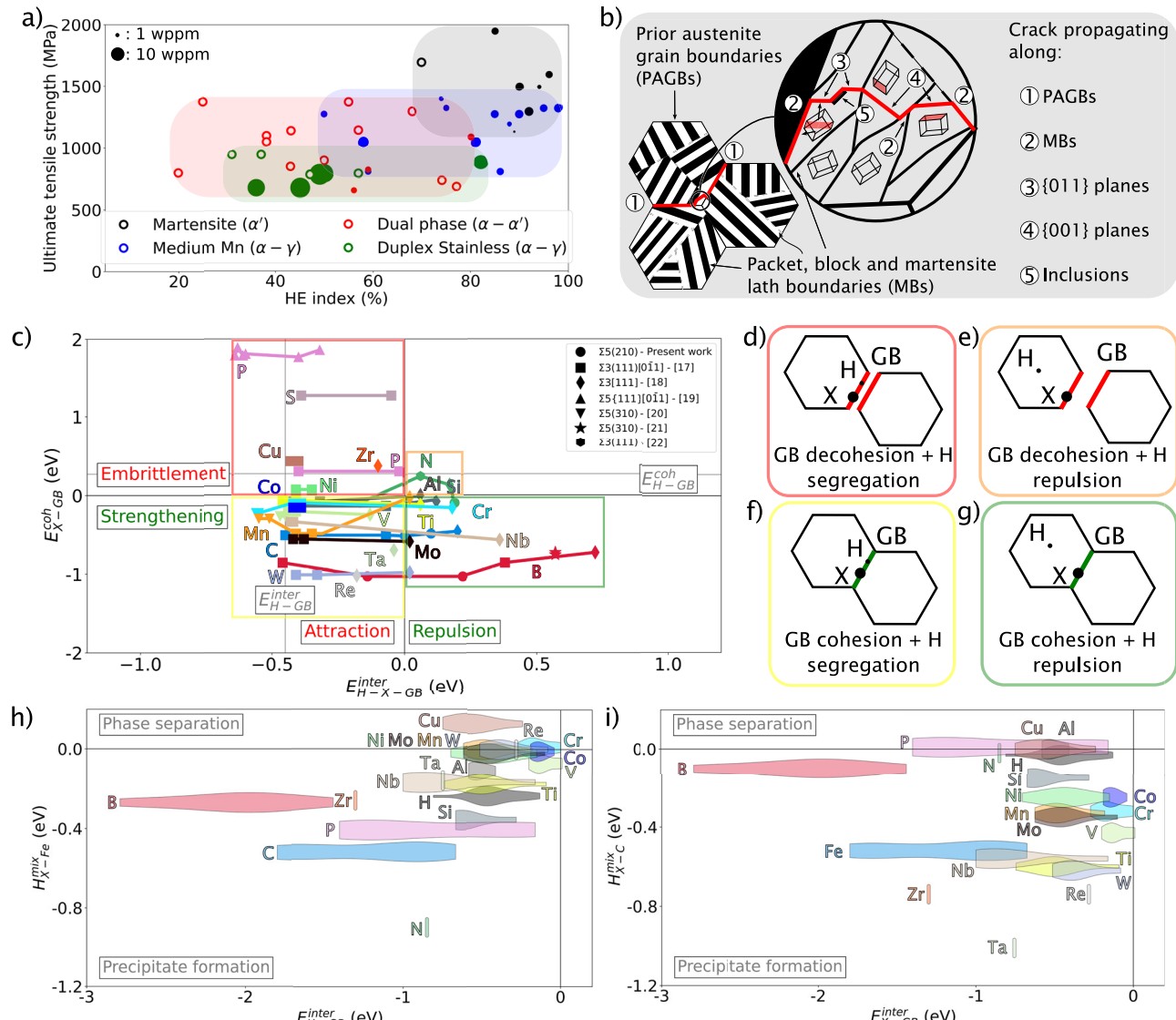

**Fig. 1 | Strategy to determine the most suited solutes to prevent hydrogen segregation at interfaces in steels. a** Ultimate tensile stress as a function of the HE index for various steel microstructures. The size of filled symbols is proportional to the hydrogen concentration, while the open ones do not provide information on the hydrogen concentration. **b** Main crack propagation observed on hydrogen-charged martensitic steels at prior austenite grain boundaries (PAGBs), {001} plane, {011} plane either from dislocation slip planes or martensite boundaries (MBs), and decohesion of inclusion/precipitate interfaces[13–16]. **c** Cohesive energy of a doping solute X with a grain boundary (GB) as a function of the interaction energy between hydrogen and the same GB doped with X[20–25]. For comparison, $E_{\text{H–GB}}^{\text{coh}} = 0.31$ eV and $E_{\text{H–GB}}^{\text{inter}} = -0.41$ eV is also represented. **d–g** Illustrations of the different mechanisms of the doping solute on the GB and H segregation. Mixing enthalpy of h) iron and **i** carbon with different solutes determined using the Miedema model[63] as a function of the interaction energy between X and GB in $\alpha$-iron ($E_{\text{X–GB}}^{\text{inter}}$). More information on the nature of GB and energy used from the literature is provided in Supplementary Note 1.

elongation of the steel grade without hydrogen, referred to as HE index. While most steel grades presented (martensitic, medium Mn, dual phase, and duplex stainless steels) are susceptible to HE, martensitic steel has higher HE indexes and is generally the most impacted. This is attributed to its high strength and complex and dense arrangement of hierarchically-arranged interfaces (Fig. 1b). This latter feature makes martensitic steel, such a compelling, widely used, low-cost, high-strength structural material. However, it also makes it particularly vulnerable to HE.

According to previous reports[13–16], HE in martensitic steels is mostly observed due to crack propagation along (i) prior austenite grain boundaries, (ii) martensite boundaries (interfaces between martensite variants that origin from one austenite grain), (iii) {001} planes (cleavage planes for body-centered cubic (BCC) crystals), (iv) {011} planes (typical slip plane family of dislocations in BCC), and (v) interface of inclusions or precipitates as illustrated in Fig. 1b.

Consequently, interfaces including grain boundaries (GB), which largely govern materials' properties with their structure and chemistry[17], are critical to the HE susceptibility of the system. Ab initio calculations can probe the likelihood of hydrogen segregation at a solute-doped GB through the interaction energy between hydrogen and the doped defect ($E_{\text{H–X–GB}}^{\text{inter}}$). Additionally, they can also probe the cohesive energy of solute-doped GB ($E_{\text{X–GB}}^{\text{coh}}$) relative to the undoped interfaces[18,19]. Although numerous works have already been developed to highlight the effect of solutes to strengthen grain boundaries, information on their effect towards hydrogen segregation remains scarce, and experimental evidence of this behavior is lacking in the literature, which is the aim of this study. While the interfacial segregation of these solutes should have a limited effect on crack propagation along crystalline planes within grains, they can avoid the decohesion of interfaces such as prior austenite and martensite boundaries.

Suggested design maps are next plotted from ab initio calculations to find the most suitable solutes to prevent hydrogen segregation at interfaces and reduce the HE susceptibility of martensitic steels. Figure 1c plots $E_{X-GB}^{coh}$, which represents the cohesive energy of a GB with different solutes X as a function of $E_{H-X-GB}^{inter}$, which corresponds to the interaction energy of hydrogen with a GB doped with a solute for different coherent site lattice of α-Fe GB[20–27]. Complementary ab initio calculations for B, C, and N on Σ5(210) α-Fe GB have been performed in this work, highlighting four specific regions depending on $E_{X-GB}^{coh}$ and $E_{H-X-GB}^{inter}$. The first region is when $E_{X-GB}^{coh} < 0$ and $E_{H-X-GB}^{inter} > 0$: it is where the best solutes against HE should be located since a strengthening of the GB with a repulsive interaction against hydrogen is observed by doping GB with X. The second one is when $E_{X-GB}^{coh} < 0$ and $E_{H-X-GB}^{inter} < 0$, where the solute X has a limited impact on HE because it strengthens GB, but hydrogen is still attracted to GB. The third one is when $E_{X-GB}^{coh} > 0$ and $E_{H-X-GB}^{inter} > 0$, where the solute X is inefficient because it has a repelling effect against hydrogen, but it embrittles GB Finally, the last region is when $E_{X-GB}^{coh} > 0$ and $E_{H-X-GB}^{inter} < 0$: it is where the solute X is detrimental due to an embrittlement of the interface in combination with an attraction of hydrogen to the GB. More information related to this figure, including the different types of GB, is provided in Supplementary Note 1. Additionally, ab initio calculations have also been performed in the present work for a range of solutes (boron, carbon, and nitrogen) in a Σ5(210)α-Fe GB with and without hydrogen. One can note that the energy $E_{H-X-GB}^{inter}$ is sensitive to the distance between hydrogen and the doping solute X within the same GB[20,21], but also to different types of GB. Consequently, it leads to an energy spectrum for one doping solute, as presented in Fig. 1c.

For the approach developed in this study, the most suited solute against hydrogen embrittlement should increase the cohesion of the GB and have a repulsive interaction with hydrogen when inserted into the crystalline defect to repel hydrogen. Guided by these metrics, boron becomes the best element because it can have a repulsive energy ($E_{H-X-GB}^{inter}$) above 0.5 eV and a cohesive energy ($E_{X-GB}^{coh}$) around −1 eV. These high hydrogen repulsion and cohesion values would provide the GB with mechanical strengthening and chemical protection against hydrogen. Niobium and carbon are also potentially good candidates due to a repulsive interaction energy between 0.2 eV and 0.5 eV at a cohesive energy around −0.5 eV. Molybdenum and tungsten would also be suited elements specifically for strengthening the GB ($E_{X-GB}^{coh}$ below −0.5 eV), but their interaction energies are at maximum close to 0 eV. Finally, aluminum, silicon, titanium, chromium, and manganese also fall within the criteria of suited protective elements against hydrogen but have a cohesive energy close to 0 eV with an interaction energy which can be at maximum at 0.2 eV.

The strategy of this work is to design interfaces that are resistant to hydrogen segregation via doping with selected solutes. It is hence crucial to evaluate the interaction between doping solutes and the matrix to avoid precipitate formation, which can be detrimental to the microstructure, even though they could also be suited for trapping hydrogen[28,29]. Therefore, Fig. 1d, e presents additional design maps plotting the mixing enthalpy of X with iron and carbon ($H_{X-(Fe, C)}^{mix}$) as a function of the interaction between X and a GB ($E_{X-GB}^{inter}$). A positive $H_{X-(Fe, C)}^{mix}$ indicates a preferential phase separation between the solute X and iron or carbon, whereas a negative $H_{X-(Fe, C)}^{mix}$ indicates a tendency to form a precipitate. We also chose to investigate the mixing enthalpy of X with carbon because it is present at a high concentration in a GB for steels. Figure 1d, e shows that all solutes are attracted to the interface (because all $E_{X-GB}^{inter}$ are negative). While solutes like boron, phosphorus, and silicon have a low mixing enthalpy ($H_{B-Fe}^{mix} = -0.37$ eV, $H_{P-Fe}^{mix} = -0.41$ eV, and $H_{Si-Fe}^{mix} = -0.36$ eV), carbon and nitrogen are the most reactive elements ($H_{C-Fe}^{mix} = -0.52$ eV and $H_{N-Fe}^{mix} = -0.90$ eV). This finding suggests that no precipitates will be formed if iron carbide and nitride formation are avoided. However, Fig. 1e shows that with carbon, niobium and tungsten are more reactive than iron to form

carbides ($H_{Nb-C}^{mix} = -0.56$ eV and $H_{W-C}^{mix} = -0.62$ eV). Consequently, boron seems to be the most suited solute against hydrogen segregation at GB in addition to carbon. Considering the strengthening effect of each solute element, molybdenum is also a good candidate, even though the repulsive energy can be at a maximum close to zero, and solutes like aluminium, silicon, chromium, and manganese should have a slightly positive effect against hydrogen segregation.

Further, we investigate the incorporation of interstitial solutes (boron and carbon) at interfaces in a low-carbon (0.15 wt% C) martensitic steel (referred to as LC hereafter). The choice of these solutes is also motivated by their fast diffusion compared to molybdenum or tungsten, which are potentially beneficial elements against hydrogen segregation, but their diffusion is too slow in martensite. In the present manuscript, we design martensitic steel with a similar microstructure, but they have drastically different behavior with hydrogen. This effect is obtained by favoring the segregation of boron and carbon at interfaces sensitive to hydrogen. Additionally, this improvement is partially due to the relaxation of the microstructure, which reduces the hydrogen uptake and avoids early deformation-induced martensite transformation, as detailed in the next section.

## Results and discussion

### Alloy design to reduce hydrogen uptake and embrittlement

Four different microstructures are investigated and are referred to as LC, LC+B, LC+LTT, and LC+B+LTT (the +B implies that boron is added, and +LTT implies that the steel is subjected to low-temperature tempering at 160 °C). While boron addition and segregation should mainly strengthen prior austenite grain boundaries, the LTT will have two impacts against HE. First, it induces carbon segregation at various interfaces. Second, it should reduce the internal stresses resulting from quenching-induced martensitic transformation. We perform electron backscattered diffraction (EBSD) and synchrotron X-ray diffraction (SXRD) analyses to characterize these four microstructures, presented in Fig. 2a–e. Previous investigations have shown that incorporating boron in steel has a minor effect on the grain size, GB distributions, and dislocation densities ($\rho$)[16,27,30]. Such effects are also observed during tempering: Fig. 2f shows a similar distribution of grain boundary types, with less than 15% of prior austenite grain boundaries for all conditions. Using the circular integration of the synchrotron experiments, we determine a similar dislocation density in martensite and austenite of $\rho^{\alpha'} = 1.2 \times 10^{15}$ m$^{-2}$ and $\rho^{\gamma} = 3.8 \times 10^{15}$ m$^{-2}$ for all microstructures (with or without boron and tempering). The dislocation density for the different systems is determined with the Williamson–Hall approach[27,31,32] and indicates that recovery or recrystallisation does not rejuvenate the microstructure in both phases when the tempering temperature is 160 °C. However, a reduction of 30% (2.2 vol% to 1.5 vol%) of the retained austenite is noted. This metastable phase should be localized at martensite lath boundaries, according to previous work on medium-carbon steel[33]. The loss in austenite volume fraction can indicate stress relaxation of the martensite, which has been observed in recent work on press-hardened steels[34]. The tempering treatment also induces a slight peak shift of both phases, which is due to the evolution of the lattice strain during treatment. Figure 2f represents the determined lattice parameter from SXRD of both phases for all microstructures. When both LC and LC+B are tempered at 160 °C for 4 h, the lattice parameter of the martensite is reduced by 0.017%, while it is reduced by 0.087% for the austenite. This lattice parameter reduction implies an increase in compressive stress in the austenite, while it implies a stress relaxation of the martensite. It is mainly related to carbon diffusion (segregating from the matrix into the phase and grain boundaries) and partition during tempering. More information is available in Supplementary Note 2 on that topic.

Additionally, it has been reported that tempering at 160 °C can induce the formation of transition carbides in both high-carbon and low-carbon steels[35,36]. These precipitates are known to trap hydrogen

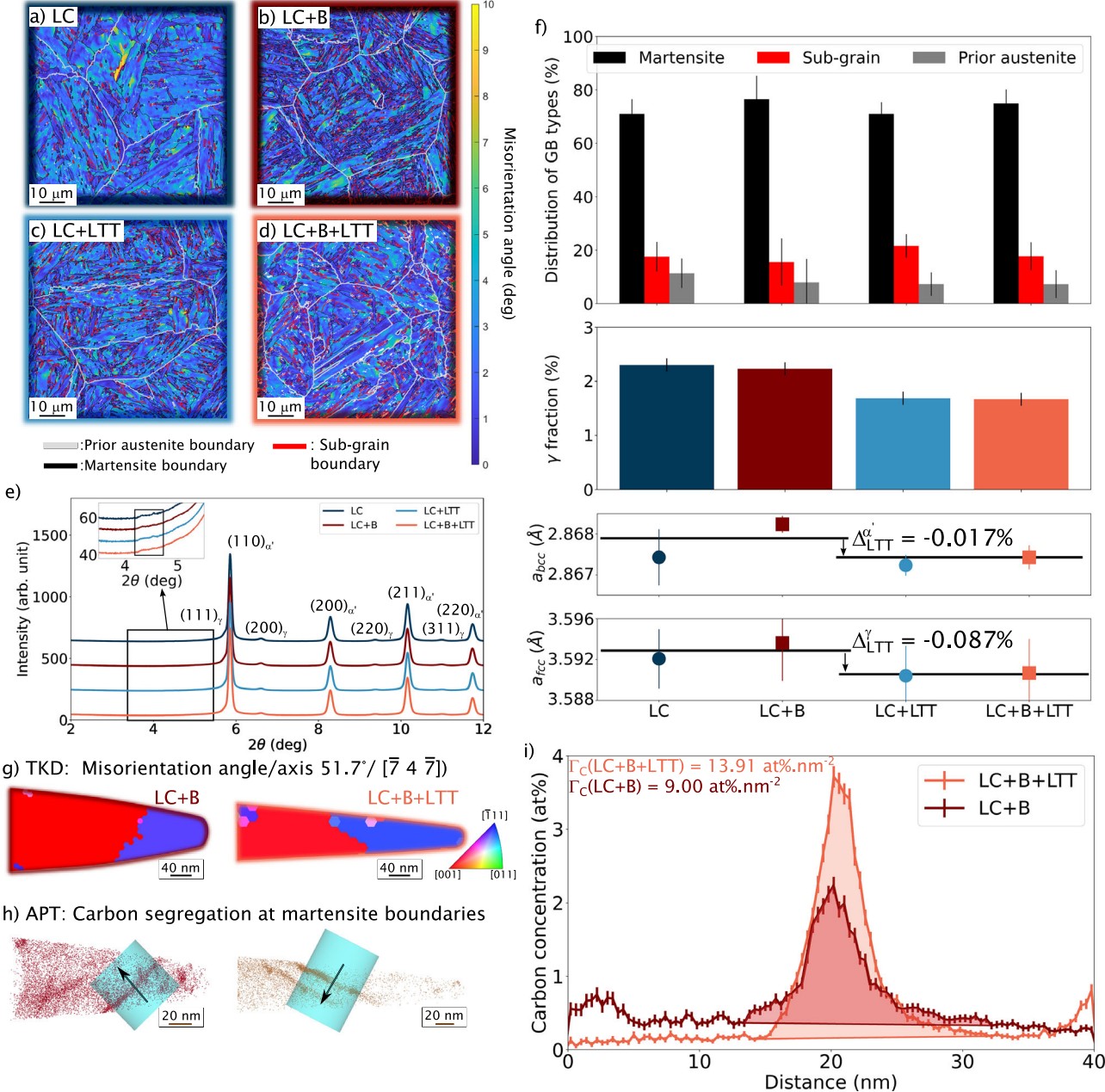

**Fig. 2 | Microstructure characterisation of boron addition and low-temperature tempering on martensitic steel.** Electron backscattered diffraction (EBSD) maps of **a** LC (low-carbon steel), **b** LC+B (boron-doped LC steel), **c** LC+LTT (tempered LC steel), **d** LC+B+LTT (boron-doped and tempered LC steel). **e** Circular integration of the 2D diffractograms from synchrotron X-ray diffraction (SXRD) measurements with an inset to highlight the contribution of carbides around 4.5°. **f** Resulting length distribution of the different boundary types, fraction of austenite, and lattice parameter measured in different phases. The error bars for the length distribution are obtained from the percentage of pixels with a confidence index higher than 0.1. The ones from the austenite fraction are deduced from the difference between the fit and the actual values of $f_\gamma$ provided in Supplementary Note 2. Finally, the ones from the lattice parameters, $a$, are deduced from the variation between the different integration peaks. The measured lattice parameter reduction of martensite and austenite is $\Delta_{LTT}^{\alpha',\gamma}$ **g** Transmission Kikuchi diffraction (TKD) measurements and **h** atom probe tomography (APT) reconstructions of a martensite boundary in LC+B and LC+B+LTT with the same misorientation angle/axis (51.7°/$[\overline{7}4\overline{7}]$)[40]. **i** Deduced 1D concentration profile and carbon excess from the APT reconstruction.

besides dislocations and grain boundaries and can significantly influence hydrogen embrittlement resistance[37,38]. Therefore, the inset of Fig. 2e highlights the contribution related to carbides (fluctuations of the diffraction pattern around 4.5°). The fluctuation is observed for all microstructures, indicating the presence of carbides for all our investigated samples. They are formed during the helium quenching, after the martensite transformation (the so-called auto-tempering), and have been described previously for LC and LC+B[27]. Further, we perform micro-hardness measurements to verify the formation of carbides[35,39].

The hardness of LC+B and LC+B+LTT are 394 ± 10 HV$_{500}$ and 397 ± 10 HV$_{500}$, respectively. The very similar hardness values suggest that the growth of these carbides or the formation of new transition carbides during the tempering is limited. Consequently, the trapping of hydrogen associated with these carbides should be similar with and without LTT. Additional information related to this topic is also provided in Supplementary Note 2.

Finally, carbon segregation at the martensite boundaries is quantified using correlative transmission Kikuchi diffraction-atom

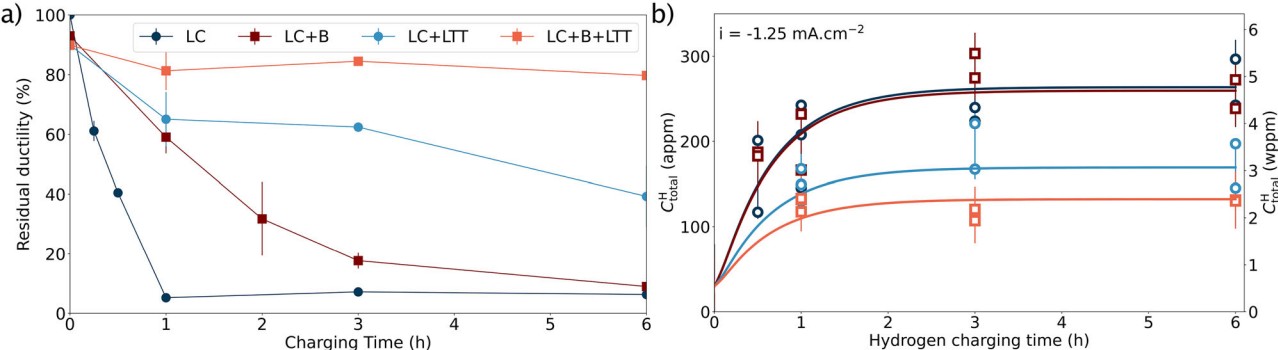

**Fig. 3 | Impact of the boron addition and tempering on the resistance against hydrogen uptake and embrittlement. a** Residual ductility (with 100% being the residual ductility of pure, untreated LC steel) and **b** corresponding hydrogen concentration for different hydrogen charging times, presented in atom part-per-million (appm) and weight part-per-million (wppm). The residual ductility is plotted corresponding to the average of two tensile test experiments, with the curve provided in the Data Source folder. The standard deviation has been determined to encompass all measured (of at least two per condition) and fitted concentrations per microstructure, resulting in a standard deviation of 0.9 wppm for LC, 0.7 wppm for LC+B and LC+LTT, and 0.6 wppm for LC+B+LTT.

probe tomography (TKD-APT) measurements. Supplementary Note 2 presents an analysis of the case of boron at prior austenite grain boundaries for steels LC and LC+B. The interfacial excess of boron is 3.3 at% nm$^{-2}$ for LC+B, while almost no boron is observed for LC[27]. Figure 2g presents the analysis for LC+B and LC+B+LTT for a martensite boundary variant with the same orientation angle/axis (51.7°/[$\overline{7}4\overline{7}$])[40]. The carbon excess at this interface, quantified by APT (Fig. 2h) and plotted in Fig. 2i, shows an increase from 9 at% nm$^{-2}$ to 14 at% nm$^{-2}$ after tempering.

The resistance against HE of these microstructures is then evaluated using slow strain rate tensile tests following electrochemical hydrogen charging. The evolution of the local strain during deformation is measured by digital imaging correlation (DIC). Figure 3.a presents the residual ductility (defined as the ratio of the total elongation of a microstructure to the total elongation of LC without hydrogen). The tensile test curves are presented in Supplementary Note 3. The corresponding hydrogen concentration ($C_{total}^{H}$) is determined by thermal desorption spectroscopy (TDS) and presented in Fig. 3b. In pure, untreated LC martensitic steel, the residual ductility is already lower than 20% after less than 1 h of charging. When boron is added, the charging time $t_H$ needs to be increased up to 3 h to observe a residual ductility lower than 20%. This strongly suggests an improved resistance against HE, mostly due to the boron segregation at prior austenite grain boundaries. This improvement has been characterized previously using correlative TKD-APT measurements[16,27] and shown in Supplementary Note 2. According to the TDS measurements, this improvement is determined for similar $C_{total}^{H}$ for LC and LC+B for all $t_H$ (saturating around 280 appm when $t_H = 2$ h). Tempering leads to a substantially lower reduction of the residual ductility across all charging times, with the residual ductility consistently above 40%. It is evidenced that the reduction of austenite fraction and carbon segregation at martensite boundaries improves the resistance against HE. Finally, the best result is observed for LC+B+LTT, where the ductility is always above 80%. It suggests a synergistic effect of boron segregation at prior austenite grain boundaries, carbon into martensite boundaries, and reduction of the austenite fraction. This improvement can be explained when determining the saturated $C_{total}^{H}$ of these microstructures. It drops to 180 appm for LC+LTT and 120 appm for LC+B+LTT. These measurements indicate an improvement in resistance against HE by a reduction in the hydrogen uptake.

Complementary experiments have also been conducted for more severe hydrogen charging conditions, still showing an improved resistance against HE of the LC+B+LTT compared to the other microstructure (Supplementary Note 3). Hydrogen permeation tests (presented in Supplementary Note 3) show a slightly reduced steady state permeation rate, suggesting a reduced hydrogen solubility due to tempering, which is also confirmed in Fig. 3b. In addition, an increase in the apparent hydrogen diffusion coefficient is observed when the steel is tempered. It might be induced by the internal stress relaxation from carbon segregation and austenite reduction, reducing the number of trapped sites for hydrogen, which is also consistent with our TDS experiments.

A detailed analysis of the crack propagation path is further conducted to understand the improved resistance to HE from boron and carbon segregations, as well as the relaxation of austenite. Previous works have shown that boron addition impedes crack propagation along prior austenite grain boundaries[13,16], due to an increase of grain boundary cohesion by a strong hybridization between the iron s, d-states, and the boron s, p-states, according to previous ab initio calculations[24,41]. Further, we opted to study secondary cracks over the study of the primary cracks because observing both sides of a crack is necessary to distinguish {011} planes within the same martensite grain or from a martensite boundary. Figure 4a-d are examples of secondary cracks induced by the presence of hydrogen in LC, LC+B, LC+LTT, and LC+B+LTT, respectively. The distribution of the crystallographic or microstructural nature of these cracks is summarized in Fig. 4e. While more than half of the cracks observed in LC+H are from prior austenite grain boundaries, this fraction drops to 20% for the material variants LC+LTT+H. It suggests an effect of carbon on reducing the crack propagation at these microstructure features in the presence of hydrogen, in agreement with previous results from the literature[42]. When boron is added, the fraction becomes lower than 10% for boron-doped variants. Considering the ab initio calculations of Fig. 1 and boron segregation previously measured through correlative TKD-APT measurements (in Supplementary Note 2). This fraction reduction underpins the tremendous improvement in HE resistance provided by boron. However, the weakest remaining microstructural features observed in materials LC+B+LTT+H are martensite boundaries and inclusions' interfaces, which fail at larger strains than the LC+H material variant. Therefore, the excess of carbon segregating at martensite boundaries only mitigates the embrittlement of these interfaces and does not completely stop hydrogen segregation at these interfaces. Further investigation is performed to determine the inclusion's type. Site-specific APT measurements, detailed in Fig. 4d, e is conducted and identify them as manganese sulfate (MnS), as shown in Fig. 4f, g. Although they can trap hydrogen (trapping energy around 0.75 eV[43]), they remain a weak point in steel, leading to facial crack initiation. Carbide precipitates can have a similar role, in addition to inclusions, as widely reported in the literature[15,43-45].

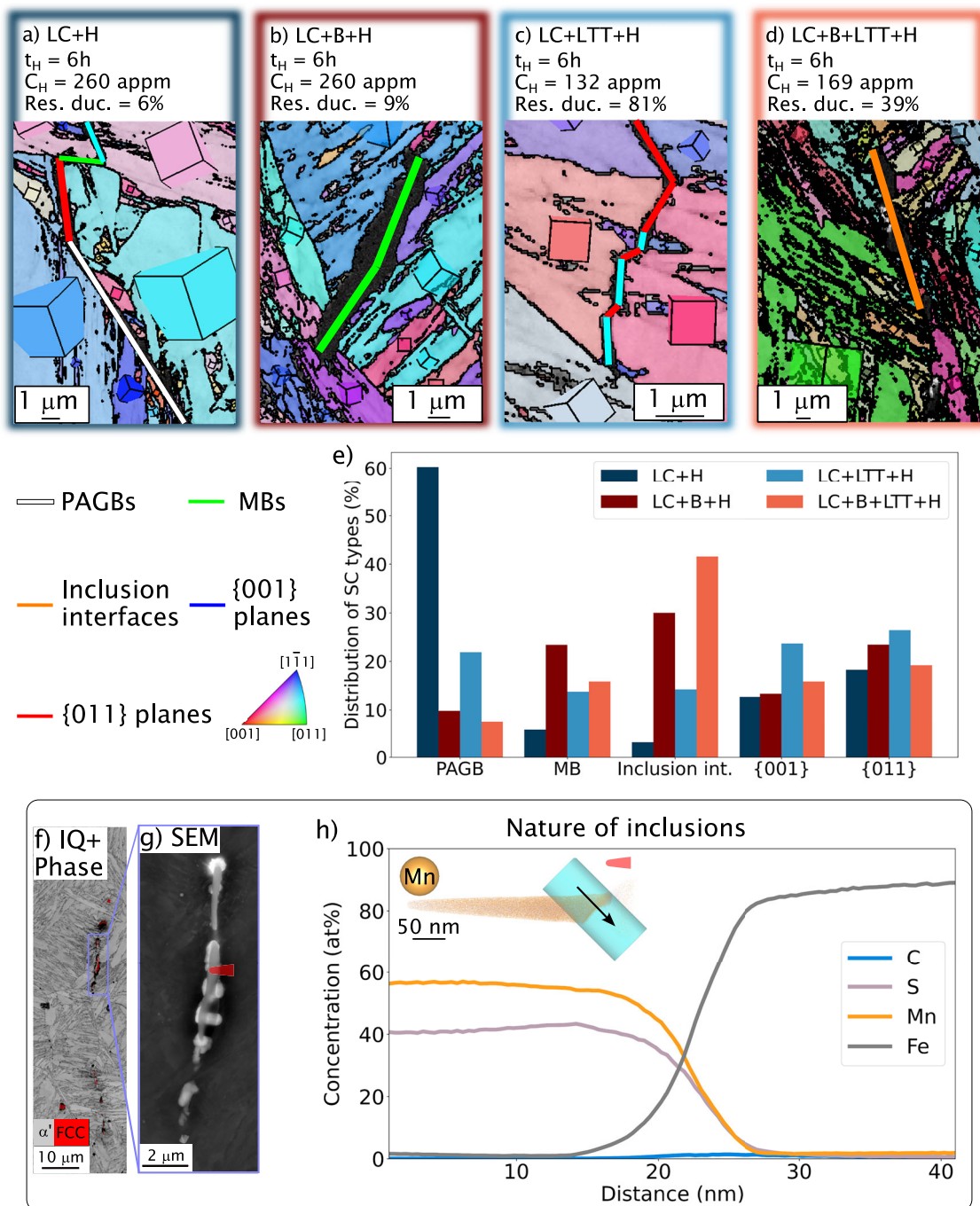

**Fig. 4 | Secondary crack analysis on hydrogen-charged ($t_H$ = 6 h) and tensile-strained samples.** Examples of secondary cracks observed in pre-charged **a** LC, **b** LC+B, **c** LC+LTT, and **d** LC+B+LTT at the subsurface of the fracture surface. The grain orientation is also represented to identify the nature of secondary cracks: (i) prior austenite grain boundaries (PAGBs), (ii) martensite boundaries (MBs), (iii) interface inclusion/matrix, (iv) {001}, and (v) {011} planes. For this analysis, indexed pixels have a mean angular deviation under 1°. **e** Distribution of different secondary cracks (SCs) natures in LC+H and LC+B+LTT+H. For all steels, the measurement is performed on more than 120 μm of cracks with images provided in Supplementary Note 4. **f** Phase map from EBSD analysis of areas with inclusions (only pixels with a confidence index higher than 0.1 are shown), with **g** image of one inclusion showing the different contrast with the matrix, and highlighting where the APT analysis is performed. **h** 1D concentration of carbon, sulfur, manganese, and iron around the interface between the inclusion and the matrix.

## Suppression of early deformation-induced martensite transformation

The existence of residual austenite is proven unambiguously by synchrotron measurements. Previous transmission electron microscopy analysis of martensite in steels with similar microstructure shows that such retained austenite might appear as an elongated phase in between martensite boundaries[33]. In the present work, some retained austenite films on martensite boundaries could be detected using EBSD by using spherical indexing. The detected films had a thickness of 20 nm and a length of 200 nm (Fig. 5a). Figure 5b plots the reconstruction of an APT analysis of this thin film of austenite. This austenite follows the curvature of the 3D reconstruction, due to the reconstruction protocol from Geiser et al. implemented in the AP Suite software and known to develop distortion at the border of the reconstruction[46]. While this approach can modify the size of the thin film, it will not change the chemical composition plotted in Fig. 5c,

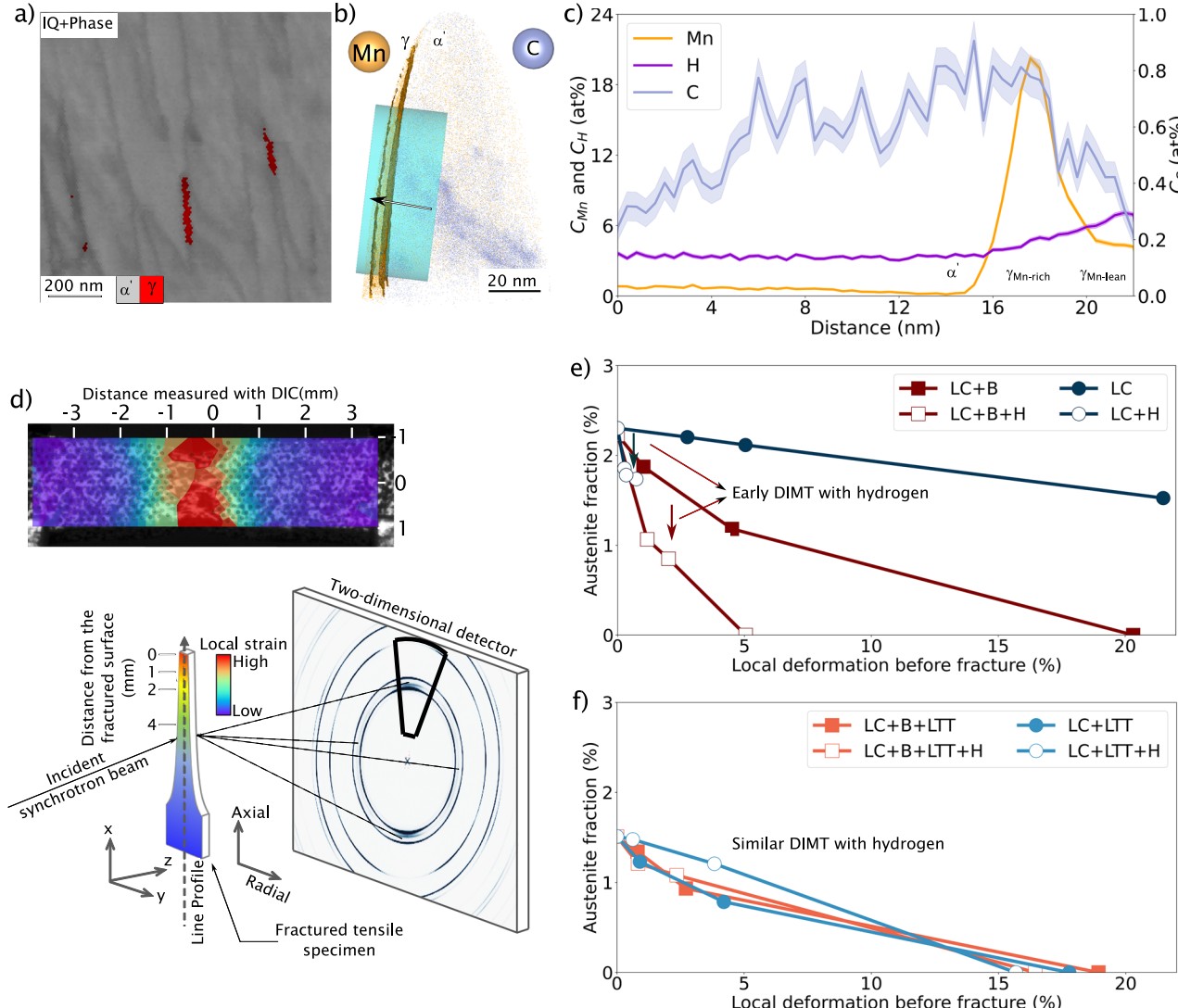

**Fig. 5 | Contribution of retained austenite at martensite boundary interfaces.** **a** Localisation of the austenite in LC+B+LTT observed by EBSD using spherical indexing and **b** APT reconstruction of the austenite with **c** Mn, C, and H concentrations. **d** Local strain of the last frame before failure of a tensile test and illustration of the synchrotron measurements: During the experiment, the fractured sample is navigated along the x direction, starting from the fracture end (highest local strain position) to the clamped region (lowest local strain position)[32]. The integration is performed only in the axial direction of the synchrotron diffraction. Fraction of austenite as a function of the local strain before fracture of a tensile strain specimen of LC and LC+B **e** without and **f** with the LTT showing the deformation-induced martensite transformation (DIMT) for each microstructure.

which shows two ranges of manganese concentration. One range, contains up to 20 at% of manganese (called $\gamma_{Mn\text{-rich}}$) and the second one $\gamma_{Mn\text{-lean}}$ with 5 at% of manganese. Previous work shows that such chemical heterogeneity improves the resistance against HE[47]. Figure 5c also shows the concentration of carbon and hydrogen. The latter solute is not pre-charged; thus, it originates from either the sample preparation or it has already been present in this phase[48,49] and can only give qualitative information on the hydrogen distribution. In the Supplementary Note 5, the evolution of the electrostatic field along the 1D concentration profile is presented using the relative charge state ratio of $Fe^+/(Fe^+ + Fe^{2+})$. The decrease of this ratio indicates an increase in the electrostatic field in the austenite, which should result in a reduction of hydrogen from residual gas ionization. Because a higher hydrogen concentration is detected in the austenite, the measured hydrogen concentration does not originate from this ionization. According to previous ab initio calculations[50], hydrogen should be more soluble in the austenite phase with a higher concentration of Mn. However, this is not observed in the APT measurements, which instead show a slightly higher concentration of hydrogen in the $\gamma_{Mn\text{-lean}}$ (more

than 6 at%) compared to the $\gamma_{Mn\text{-rich}}$ phase (between 4 at% and 5 at%). One possible explanation is the higher concentration of carbon (close to 0.8 at%) observed in the martensite and the $\gamma_{Mn\text{-rich}}$ phase, as the steel is tempered at 160 °C while it is closer to 0.2 at%C. in the matrix and the $\gamma_{Mn\text{-lean}}$ phase. Additionally, one can note that a lower carbon concentration is observed after tempering in the $\gamma_{Mn\text{-lean}}$ phase even though carbon partitioning should have occurred. This result is obtained because of the carbon segregation at the grain boundary next to the $\gamma$ phase, which is quantified in Supplementary Note 5. Assuming a repulsive interaction between carbon and hydrogen (which can be observed through ab initio calculations for GB in $\alpha$-iron and seen at prior austenite grain boundary[42]), carbon could slow down hydrogen diffusion into this phase by increasing its concentration at the interface.

Finally, synchrotron X-ray diffraction measurements is performed at a defined distance from the fracture surface on deformed samples, both uncharged and precharged with hydrogen for 3 h. These measurements aim to better understand the stability of the retained austenite as a function of the retained austenite after treatment and with

hydrogen. The local strain is determined using the last digital imaging correlation frame before fracture (Fig. 5d). From these post-mortem analyses, the fraction of austenite is plotted in Fig. 5e, f for untempered and tempered conditions, respectively. Deformation-induced martensite transformation is observed when the local strain is increased, a phenomenon that is more important for LC+B than LC (Fig. 5e). When hydrogen is incorporated into both microstructures, the transformation is observed for lower strain levels (Fig. 5e). It indicates a hydrogen-induced reduction in austenite mechanical stability. This earlier martensite formation can induce cracking and is directly linked to the susceptibility of HE[51]. However, this change in austenite stability due to hydrogen is not observed when the microstructure is relaxed after tempering (see LC+LTT and LC+B+LTT in Fig. 5f). It suggests that either hydrogen ingress or trapping is lower within austenite of tempered steels or/and the austenite is becoming more stable after tempering.

Since no early deformation-induced martensite transformation is seen for hydrogen-precharged tempered steel, the retained austenite is either more stable (due to stress relaxation) or/and the hydrogen uptake is reduced in this phase (possibly due to carbon segregation). To clarify this point, we determined the equivalent hydrostatic stress ($\sigma_{\text{hydro}}^{\alpha',\gamma}$) from the lattice parameter reduction ($\Delta_{LTT}^{\alpha',\gamma}$), using[52]:

$$\sigma_{\text{hydro}}^{\alpha',\gamma} = \frac{E^{\alpha',\gamma}}{1-2\nu}\Delta_{LTT}^{\alpha',\gamma}. \tag{1}$$

with $E^{\alpha',\gamma}$ the isotropic Young's modulus of martensite and austenite ($E^{\alpha'}$ = 180 GPa and $E^{\gamma}$ = 172 GPa) and $\nu$ the Poisson's ratio ($\nu$ = 0.3)[52]. Figure 2 shows that the tempering induces a slight reduction of the lattice parameter for both martensite ($\Delta_{LTT}^{\alpha'}$ = −0.017%) and austenite ($\Delta_{LTT}^{\gamma}$ = −0.087%) from carbon partitioning (loss of carbon in solid solution from the martensite) and a reduction of the austenite (due to stress relaxation). It leads to a relaxation of −76 MPa in the martensite ($\sigma_{\text{hydro}}^{\alpha'}$), and an increase of 374 MPa of the compressive stress in the austenite ($\sigma_{\text{hydro}}^{\gamma}$ = −374 MPa). In Supplementary Note 5, we have conducted ring core experiments on LC and LC+LTT, measuring a similar total hydrostatic stress. It strongly suggests that the tempering induces stress relaxation, as discussed in previous work[34], stabilizing the austenite. The resulting reduction of hydrogen ingress from stress relaxation in both phases ($\Delta S_H^{\alpha',\gamma}$) can be theoretically estimated using[53]:

$$\Delta S_H^{\alpha',\gamma} = \exp\left(\frac{\sigma_{\text{hydro}}^{\alpha',\gamma}\bar{V}_H^{\alpha',\gamma}}{k_{\text{B}}T}\right) \tag{2}$$

With $\bar{V}_H^{\alpha',\gamma}$ the partial volume of hydrogen in martensite and austenite ($\bar{V}_H^{\alpha'}$ = 3.3 Å$^3$/at.[54] and $\bar{V}_H^{\gamma}$ = 2.0 Å$^3$/at.[55]), $k_{\text{B}}$ the Boltzmann constant and $T$ the temperature. The resulting solubility reduction in martensite and austenite are $\Delta S_H^{\alpha'}$ is 94% and $\Delta S_H^{\gamma}$ is 83%. However, Fig. 3b shows that the hydrogen concentration of the tempered LC is 60% for LC and 50% for LC+B, which can be explained by the reduction of hydrogen ingress from boron and carbon segregation at interfaces.

To summarize, atomistic calculations suggest that interstitial solutes, particularly boron and carbon at grain boundaries, have a tremendous effect against hydrogen segregation due to their capacity to strengthen interfaces and a repulsive interaction concerning hydrogen. While solute strengthening of interfaces in steel has been well documented in the literature, the influence of solute strengthening on hydrogen segregation has been less explored (some examples have been detailed in the design treasure maps of Fig. 1c). In the present work, we successfully demonstrate experimentally how these insights can be used to design martensitic steels, a grade that exhibits a high HE susceptibility. We showcase a significant improvement in the resistance against HE, with a reduction of the hydrogen ingress, due to interstitial solute segregation at interfaces (GB and retained austenite/

martensite) and stress relaxation of the retained austenite. Our comprehensive study over multiple scales finally provides direction for further optimization of the steel, targeting strengthening of martensite boundaries and limiting inclusions and precipitates, to go beyond this already unprecedented improvement in HE resistance.

## Methods

### Ab initio calculations

The calculations are carried out using the projector augmented wave (PAW) potentials as implemented in Vienna Ab initio Simulation Package (VASP) v5.4.4 code[56–58]. The calculations' details are identical to those presented in previous work to determine the interaction energy between hydrogen and a Σ5(210) $\alpha$-Fe GB at 0 K[27]. The cohesive energy of a solute doped grain boundary ($E_{X-GB}^{coh}$) is obtained by using the formulation from the work of Rice and Wang[18]:

$$E_{X-GB}^{coh} = \frac{(E_{X-GB}^{SC} - E_{GB}^{SC}) - (E_{X-FS}^{SC} - E_{FS}^{SC})}{N_X} \tag{3}$$

with $E_{X-FS}^{SC}$ and $E_{FS}^{SC}$ supercells containing free surface with and without the doping solute X, $N_X$ the number of doping solute incorporated at the interface, and $E_{X-GB}^{SC}$ and $E_{GB}^{SC}$, supercells containing GB with and without X, respectively. Following this convention, positive $E_{X-GB}^{coh}$ implies embrittlement because X prefers to segregate into free surfaces than GBs and negative $E_{X-GB}^{coh}$ implies a strengthening of GB with X.

This energy is plotted as a function of the interaction energy between hydrogen and a solute-doped GB ($E_{H-X-GB}^{inter}$). It describes the capacity of hydrogen to segregate at that crystalline defect when it is doped with X and it is defined as:

$$E_{H-X-GB}^{inter} = (E_{H-X-GB}^{SC} - E_{X-GB}^{SC}) - (E_{H-bulk}^{SC} - E_{bulk}^{SC}) \tag{4}$$

with $E_{H-X-GB}^{SC}$, the energy of the supercell containing hydrogen in a solute-doped GB. $E_{H-bulk}^{SC}$ and $E_{bulk}^{SC}$ are the energies of the supercell of a bulk $\alpha$-Fe with hydrogen in tetrahedral site, and without solute, respectively.

The HE index presented in Fig. 1 is determined using:

$$HEI = \frac{\varepsilon^0 - \varepsilon^H}{\varepsilon^0} \tag{5}$$

with $\varepsilon^0$ and $\varepsilon^H$ the total elongation of a system without and with hydrogen, respectively.

### Materials

The chemical compositions of both low-carbon steel without and with B (LC and B+LC, respectively) were Fe-0.15C-1.5Mn-0.0005B and Fe-0.15C-1.5Mn-0.0024B (wt%), respectively[16]. All steels were homogenized following the procedure explained previously[27] and were next heat treated in a Bähr DIL805 dilatometer with an austenitization temperature of 1373 K for 30 s, and then quenched using helium gas (with a cooling rate of 230 K.s$^{-1}$). Then, the low-temperature tempering is performed at 433 K for 4 h in the same apparatus after quenching LC and B+LC.

### Electron back-scattered diffraction analysis

The EBSD maps of Figs. 2, 4f, and 5a have been acquired using a high-resolution field emission GEMINI SEM 450 (Carl Zeiss Microscopy) equipped with a Velocity detector. An acceleration voltage of 15 kV, probe current of 5 nA, and step size of 0.1 μm have been chosen for the mappings. The post-processing of all images has been carried out using the TSL OIM Analysis v8 on pixels with a confidence index above 0.1 on a surface of at least $1.5 \times 10^{-4}$ μm$^2$. Then, the MTEX 5.11.1 software toolbox based on MATLAB R2021b was used to identify the

misorientation angle and axis of all grain boundaries to reconstruct prior austenite grain boundaries from the EBSD maps[59]. Grain boundaries were separated into three types: (i) Martensite boundaries that are grain boundary variants formed with the Kurdjumov–Sachs orientation relationship, (ii) Prior austenite grain boundaries that are other random boundaries with a misorientation angle larger than 8°, and (iii) Sub-grain boundaries, which are low-angle grain boundaries, usually representing martensite lath boundaries[27]. Spherical indexing was used to analyze EBSD patterns that were recorded with a 10 nm step size at a low acceleration voltage of 10 kV. This combination of measurement parameters improves the spatial resolution because of the high robustness of pattern analysis, which shows signals from several grains. This allowed us to detect some of the larger austenite lamellae in steel. The presence of a Kurdjumov-Sachs orientation relationship between the determined austenite and the neighboring martensite proved the austenite's correct detection. Then, the crack analysis has been conducted on EBSD maps acquired using a Helios 5 FFIB (ThermoFischer) equipped with a CMOS detector. An acceleration voltage of 15 kV, probe current of 6.2 nA, and step size of 0.05 μm have been chosen for the mappings. Then, the MTEX 6.2.beta.3 software toolbox based on MATLAB R2023b was used to identify the type of secondary cracks on more than 120 μm of crack for each sample.

### Synchrotron X-ray diffraction measurements

The microstructure has been analyzed using synchrotron X-ray diffraction measurements. They were conducted at Deutsches Elektronen-Synchrotron (DESY, Hamburg, Germany) on the Petra III P-02.1 beamline at 60 keV. A high-energy transmission X-ray beam with a wavelength of 0.207381 Å was shed on square-shaped specimens ($10 \times 10 \times 1$ mm³) to collect two-dimensional diffractograms at a working distance of 969 mm. All specimens were individually heat-treated in Bähr DIL805 dilatometer, and aged for one month at room temperature before performing the SXRD measurements on each specimen. Before conducting quantitative diffraction analyses, the instrumental parameters have been calibrated using the diffraction patterns of NIST standard $LaB_6$. All recorded two-dimensional diffractograms have been post-processed using the GSAS-II v5.2.0 software[60]. The fraction of austenite has been determined by integrating peaks from both austenite and martensite, and the dislocation density $\rho$ in both phases has been determined using the Williamson-Hall approach[31] on the different tempered specimens:

$$\beta \cos(\theta_{hkl}) = \frac{\lambda}{D} + \varepsilon_{\text{micro}} \sin(\theta_{hkl}) \qquad (6)$$

with $\beta$ the full width at half maximum of the diffraction peak at $\theta_{hkl}$ (the position of the hkl reflection group), $\lambda$ the wavelength of the beam (0.207381 Å), $\varepsilon_{\text{micro}}$ the micro-strain. The dislocation density $\rho$ can then be estimated using[31]:

$$\rho = \frac{2\sqrt{3}\varepsilon_{\text{micro}}}{bD} \qquad (7)$$

with $b$ being the magnitude of the Burgers vector of the screw and edge dislocations in the martensite (2.48 Å) and in the austenite phases (2.54 Å), respectively.

### Transmission Kikuchi diffraction

Prior to conducting the APT measurements, specimens were analyzed using TKD to characterize the grain boundaries with a Digiview V) EBSD detector on MERLIN SEM (Zeiss Microscopy) and were operated with an acceleration voltage of 30 kV and a probe current of 2 nA.

### Atom probe tomography

Samples have been prepared using a dual-beam SEM-focused ion beam (FIB) instrument (FEI Helios Nanolab 600i) using an in situ lift-out procedure. The prepared APT specimens were investigated using a CAMECA LEAP 5076XS instrument, operating in laser mode at 60 K with a pulse rate of 200 kHz, pulse energy of 30 pJ, and a detection rate of 50 ions per 1000 pulses. The three-dimensional reconstructions have been performed using the AP Suite 6.3 software.

### Tensile test experiments

Slow strain rate tensile testing was conducted on a Kammrath & Weiss test stage coupled with the digital image correlation (DIC) technique, and the data were processed with the ARAMIS Professional 2020 software. Tensile specimens with a gauge length of 4 mm and a width of 2 mm were used, and the tests were performed at a strain rate of $7.5 \times 10^{-5}$ s⁻¹. At least two samples were tested for each hydrogen-charged condition. The DIC technique determined the engineering strain and local strain distribution.

### Hardness measurements

Micro-hardness measurements were performed using an HZ50-4 device with a Vickers diamond indenter from Presi at room temperature. 10 indents with a load of 500 g. and a dwell time of 10 s has been performed for each microstructure.

### Hydrogen charging condition and concentration measurements

Hydrogen was incorporated electrochemically into both steels using a three-electrode system. The procedure consisted of imposing a current density in a 0.1 M $H_2SO_4$ aqueous solution with 0.3% of $NH_4SCN$. For all conditions, the time between the end of hydrogen charging and the beginning of the tensile test experiments was 15 min. However, the time between the end of hydrogen charging and the melting experiments was less than 5 min. The hydrogen concentration was determined by the melt extraction method using a thermal conductivity detector from the G8 GALILEO apparatus of Bruker®. The measurements were performed on samples with a dimension of $5 \times 5 \times 1$ mm³. The equation used to follow the total hydrogen concentration in different microstructures in Fig. 3 is[61]:

$$C(t) = \frac{4C_l}{\pi} \int_{x=-h/2}^{x=h/2} \sum_{n=0}^{\infty} \frac{(-1)^n}{(2n+1)} \cos\frac{(2n+1)\pi x}{h} \left(1 - \exp\frac{\pi^2(2n+1)^2 D_H t}{h^2}\right) dx \qquad (8)$$

with $h$ and $D_H$ the thickness and the hydrogen diffusion coefficient in martensite ($D_H = 4.5 \times 10^{-11}$ m² s⁻¹[62]). The hydrogen concentration $C_l$ is the limit fitted to the hydrogen concentration determined by thermal desorption spectroscopy.

## Data availability

The raw data generated in this study, required to reproduce these findings, have been deposited in the public community repository Figshare: https://doi.org/10.6084/m9.Figshare.30086674.

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

## Acknowledgements
The authors thank V.S. Razumovskiy, J. Macchi, F. Barbe, and R. Henry for fruitful discussions. The help of M. Adamek, C. Bross, K. Angenendt, A. Laimmer for technical support. Synchrotron X-ray diffraction experiments have been conducted on Beamline P02.1, PETRA III at DESY (proposal No. I-20230183, I-20231121, i-20240072), with technical support from Dr. Alba San Jose Mendez, Dr. Alexander Schökel, and Martin Aashov Karlsen. G.H. and S.W. acknowledge the financial support from the Alexander von Humboldt Foundation through grant numbers 3.3-FRA-1227460-HFST-E. and n°3.1-USA-1237011-HFST-P, respectively.

## Author contributions
G.H., D.P., and D.R. secured funding. G.H. and S.W. conceived the presented idea. D.P. and D.R. supervised the project. A.T. performed the atomistic simulations. G.H. conducted the microstructure characterisation through EBSD, TKD, and APT, and performed the tensile test and hardness experiments. G.H. and S.Z. conducted the EBSD analysis with spherical indexing. G.H., S.W., and X.D. performed the SXRD experiments. G.H. and X.D. performed the TDS measurements. J.L. and A.Z. did the ring core and permeation experiments, respectively. B.S., S.Z., B.G., D.P., and D.R. contributed to the data analysis. G.H. wrote the original paper. B.S., S.Z., B.G., D.P., and D.R. revised the paper.

## Funding

## Competing interests
The authors declare no competing interests.
