## [Transparent Peer Review file · Nature Communications]

Protection of metal interfaces against hydrogen-assisted cracking

Corresponding Author: Dr Guillaume Hachet

Version 0:

Reviewer comments:

Reviewer #1

(Remarks to the Author)

This manuscript investigates hydrogen embrittlement in martensitic steel and proposes potential mitigation strategies. The following issues must be addressed before the paper can be considered for acceptance:

1. The curve fitting accuracy in Figure 3b is not high, especially when the hydrogen filling time increases but the hydrogen content decreases multiple times. It is suggested to consider the factor of experimental error and repeat the experiment
2. On page 9, Figure 4 compares the crack details of LC+H and LC+B+LTT+H to illustrate that the addition of element B resists hydrogen embrittlement. However, the variables here are not unique (LTT and B). If one wants to prove the effect of the addition of element B alone, the variable should be kept unique, such as by comparing LC+H with LC+B+H, or by comparing LC+LTT+H with LC+B+LTT+H
3. While the suppression of prior austenite grain boundary cracking in LC+B+LTT is evident (Fig. 4c), the specific roles of boron/carbon segregation and stress relaxation in this process remain unclear.
4. The discussion on precipitates in the text is relatively simple. Please explain why the improvement of hydrogen embrittlement performance is not related to the hydrogen trap precipitates formed during tempering
5. The content of elements in martensitic and austenite is given in Figure 5c. Theoretically, carbon partitioning from martensite to austenite is expected to occur during low-temperature tempering. However, in Figure 5c, there is almost no difference in the C content between the austenite and martensite phases. Please check the accuracy of the data or provide an explanation.
6. Figures 5e and 5f compare the transformation behavior of retained austenite with and without tempering before and after hydrogen charging, leading to the conclusion that "suggesting that hydrogen ingress is lower at martensite/austenite interfaces for tempered steels." However, this observation could also be explained by the increased stability of retained austenite during low-temperature tempering (a well-documented phenomenon in the literature), which consequently makes it more resistant to transformation under hydrogen charging. It is recommended that the authors provide a more comprehensive discussion regarding the behavior of retained austenite.
7. Given that existing literature has well documented the beneficial effects of certain elements (e.g., C) segregating at interfaces—enhancing interfacial cohesion energy and inhibiting the decohesion (regardless of hydrogen charging)—the authors should more prominently highlight the novel aspects of their work in this context.
8. There are many writing flaws that make this article less readable : When an abbreviation appears for the first time in the main text, the complete term should be given, such as "GB" in the third paragraph; Each symbol in the formula should be given an explanation, such as NX in Formula (1); In the comments below Formula (1), two ESC X-FS are repeated, etc.

Reviewer #2

(Remarks to the Author)

The authors present a systematic study that demonstrates a design strategy to mitigate hydrogen embrittlement, by using model steel systems and advanced materials characterisation techniques. Although the beneficial effects of B and C on hydrogen embrittlement have been proposed and reported previously, the topic remains of high interest due to its potential impact, attracting attention from both the research community and industry.

In this study, guided by a literature review and DFT simulations, the authors show that B and C segregation at grain boundaries (GBs) can effectively mitigate hydrogen embrittlement. To explore this, they designed four types of model steels with different B and C segregation states at GBs. Microstructural analysis reveals only minor differences across the samples, aside from variations in the proportion and dislocation density of austenite grains.

The authors conducted slow strain rate tensile tests and thermal desorption spectroscopy (TDS) to assess hydrogen susceptibility and concentration. EBSD analysis of fractured samples shows that the inclusion of B and C significantly reduces cracking at prior austenite grain boundaries. After further consideration of the roles of inclusions and retained austenite, the authors conclude that B and C segregation at GBs plays a critical role in mitigating hydrogen embrittlement. This could be a very effective design strategy to mitigate hydrogen embrittlement in martensitic steels.

I recommend revision to address the following comments:

In Fig. 1c, the energy spectrum of B data exhibits a wide spread, similar to several other elements in the 'strengthening' scheme. Besides the distance between B and the solute atom (discussed on page 3), could the type of GB also be a factor? The authors have listed GB information in the Supplementary Data. Would it be helpful to incorporate GB type into the plot?

Fig. 3b shows that the saturated hydrogen concentration nearly halves after LLT. Could the authors clarify this observation? Changes in retained austenite or dislocation density seem insufficient to explain such a large difference. Will it be possible to further analyse the TDS data and draw useful conclusions from the perspective of quantifying trapping energies, and connect H content with microstructural feature.

I assume the EBSD was performed over a large area to improve the reliability of fracture statistics. However, the quality of the maps in Fig. 4 could be improved. Reducing the pixel size would enable more detailed analysis.

In the APT dataset, the shape of the austenite phase follows the curvature of the reconstructed 3D model. This raises questions about the reconstruction approach. In addition to the introduction of hydrogen during sample preparation and residual hydrogen, 1Da signals can also originate from environmental hydrogen in the APT chamber. Given the low content and surface-biased distribution, it is difficult to extract meaningful insights about hydrogen trapping. The authors are advised to interpret the hydrogen signal with more caution.

In the caption of Fig. 5, the phrase "local strain before deformation" may be a typo and should perhaps read "local deformation before fracture."

In the Supplementary Data, should Tables 6 and 7 appear back-to-back? They are currently separated by Part 2.

In Fig. 5e, the in-situ synchrotron results show much more pronounced martensitic transformation in the H-charged, post-LTT sample. Could the authors elaborate on this observation?

Reviewer #3

(Remarks to the Author)

The manuscript titled "Protection of metal interfaces against hydrogen-assisted cracking" written by Guillaume Hachet et al. explores the role of boron and carbon at interfaces in martensitic microstructures in enhancing resistance to hydrogen embrittlement. The topic is both fascinating and highly relevant to the scope of Nature Communications. The manuscript is well written and demonstrates a high degree of technical completion. Furthermore, the findings are expected to have a significant impact on steel applications across various industrial fields. However, there are several aspects of the manuscript that require further clarification and revision in order to enhance the persuasiveness of the author's arguments. Additionally, some modifications are needed to improve the overall readability of the text. Therefore, I believe that a number of revisions are necessary prior to publication, and a Major revision is recommended to improve the completeness and clarity of the manuscript. I would like to point out the following major issues that need to be addressed:

1) Please ensure that the full names of all abbreviations (e.g., HEDE, HELP, and so on) are clearly stated upon first use in manuscript.

2) A more accessible explanation of the Interaction energy ($GB(E_{(H-X-GB)^{inter}})$) and cohesive energy ($GB(E_{(H-GB)^{coh}})$) would be helpful for readers who are not familiar with these concepts. While a detailed description appears to be included in the first part of the Supplementary information, a brief explanation should be provided in the main text, along with a clear reference directing readers to Supplementary Section 1 for further details.

3) In the explanation of Fig. 1c, the authors state: "The GB energy design map reveals that boron, carbon, molybdenum, tungsten, niobium, and chromium are the best elements because they increase the cohesion of GB and can have a repulsive interaction with hydrogen." However, several of these elements appears to fall at least partially within the attraction region on the map. It would be helpful if the authors could provide additional clarification on the criteria used to make this assessment.

4) Figures 1d-e would benefit from a more reader-friendly explanation. For instance, it could be helpful to explicitly state that a more negative mixing enthalpy implies a stronger tendency for bonding. Additionally, the statement that “carbon and nitrogen are the most reactive elements with iron” appears somewhat ambiguous. Phosphorus, which is also an interstitial element, shows a similar trend to carbon, making the distinction unclear. Furthermore, the claim that “tungsten and niobium are even most reactive than iron to form carbide” seems inconsistent with the behavior of other elements such as Zr, Re, Ta, and Ti, which also appear highly reactive based on the data, yet are not discussed. Finally, the concluding remark that “boron is the most suited solute against hydrogen segregation at GB, in addition to molybdenum and carbon” feels somewhat abrupt and unsubstantiated given the preceding analysis. A more coherent and supported interpretation is recommended.

5) In the initial description of the experimental procedures, the purpose of the low temperature tempering (LTT) treatment should be clearly articulated. Since readers are likely to focus on the role of boron and carbon from the beginning of the manuscript, the sudden introduction of LTT may seem abrupt or confusing. It is therefore recommended that the rationale behind the LTT treatment be explicitly stated at the point where it is first mentioned, to provide better context and improve the logical flow.

6) In Fig. 2f, the method used to determine the distribution of grain boundary (GB) types should be described more precisely. Additionally, the authors should briefly explain the reason behind the observed decrease in austenite fraction after the LTT treatment. Moreover, it is important to clarify how the dislocation densities of martensite and austenite phases were individually measured or calculated for each specimen. If a detailed explanation is not possible within the main text, at the very least, the authors should provide a citation to a reference that employed the same or a similar methodology.

7) The sentence “This effect is mostly due to the presence of boron in prior austenite boundaries” would benefit from a clear reference to Supplementary Fig. 2, as it appears to provide supporting evidence for this claim. Explicitly directing the reader to the relevant figure will help strengthen the author’s argument and improve clarity.

8) In Fig. 3c, the point-curve graph presents data with a relatively wide range indicated around the curves. It is important to clarify what this range represents – whether it reflects experimental variability, standard deviation, confidence intervals, or some other metric. Providing this explanation will help readers interpret the data more accurately.

9) The statement “Previous work has shown that a boron addition impedes crack propagation along prior austenite boundaries, as depicted in Fig. 1.b [9, 12].” Would benefit from a brief explanation of the underlying mechanism. Even a concise summary of how boron contributes to impeding crack propagation – such as through grain boundary segregation, strengthening, or chemical interaction – would help readers better understand the context and significance of this finding.

10) Despite identical electrocharging conditions before and after LTT, there appears to be a substantial difference in hydrogen ingress. While it is reasonable to expect that hydrogen embrittlement sensitivity can vary due to changes in hydrogen trapping sites, it is more difficult to understand why the total amount of hydrogen ingress itself would differ significantly under the same charging conditions for similar steel-based specimens. This discrepancy requires clear explanation. Including permeation test results would substantially strengthen the author’s argument; if such data are unavailable, a more detailed rationale should still be provided to justify this observation.

11) The LTT condition employed in this study – 160 oC for 4 hours – is known to allow the formation of epsilon (ϵ) -carbide in martensitic structures. Since ϵ -carbide are generally regarded as relatively strong hydrogen trapping sites, they could significantly influence hydrogen embrittlement resistance. However, the manuscript does not provide any discussion or experimental analysis related to carbide precipitation before and after LTT. Given that strong hydrogen trapping sites such as ϵ -carbide can significantly influence hydrogen embrittlement resistance, it is essential not only to investigate the carbide evolution before and after LTT, but also to explicitly consider how such carbide formations may alter the overall resistance to hydrogen embrittlement. Experimental data or at least a discussion on this aspect is essential to support the interpretation of hydrogen-related behavior. Relevant references include: ① Transition from carbon clusters to ϵ , θ -carbides in a quenched and aged low-carbon ferritic steel, and ② Atomic-scale observation of hydrogen trap sites in bainite–austenite dual-phase steel by APT.

12) In Fig. 4c, the authors present results only for the LC+H and LC+B+LTT+H specimens. It would be helpful to explain why these two specific samples were selected for analysis. If data are available for all four specimen conditions, providing the distribution of SC classifications for the full set would offer a more comprehensive understanding and strengthen the overall argument.

13) It is recommended that MnS in Figs. 4d-f be explicitly labeled as an inclusion, as this terminology more accurately reflects its metallurgical nature.

14) For Fig. 5c, it is recommended – if feasible – to include data showing changes in hydrogen trapping behavior after hydrogen charging. Such information would greatly enhance the credibility and depth of the analysis, providing stronger support for the author’s interpretation.

15) In Figs. 5e-f, the data suggest that deformation-induced martensitic transformation in austenite is significantly suppressed after LTT, indicating that the austenite has been stabilized. It would be beneficial to include a brief explanation of the possible mechanisms behind this stabilization, such as carbon enrichment, changes in dislocation density, or retained austenite morphology.

Reviewer #4

(Remarks to the Author)

Version 1:

Reviewer comments:

Reviewer #1

(Remarks to the Author)

Thank you to the authors for their thorough and thoughtful responses. All concerns have been adequately addressed.

Reviewer #2

(Remarks to the Author)

The authors did a good job addressing all my comments. The quality of the paper has been significantly improved.

Reviewer #3

(Remarks to the Author)

The authors have responded to the reviewer's comments in a thorough and sincere manner. The explanations provided are reasonable and generally acceptable from a scientific standpoint. Based on the revisions and clarifications made, I find the manuscript to be acceptable for publication.

Reviewer #4

(Remarks to the Author)

We cordially thank the Editor and the Reviewers for the time and effort taken to evaluate our manuscript. Below, please kindly find detailed answers to all the comments, suggestions, and questions. Changes are highlighted in red in the revised manuscript.

Reviewer #1

Reviewer's comment: 1. The curve fitting accuracy in Figure 3b is not high, especially when the hydrogen filling time increases, but the hydrogen content decreases multiple times. It is suggested to consider the factor of experimental error and repeat the experiment.

Response: We agree with the reviewer that the standard deviation of the experiment is relatively large compared to the measured hydrogen concentration. Such an effect is observed mainly because the solubility of hydrogen is low (there is always less than 6 wppm of H in the different microstructures, and the standard deviation of the measurement is between 0.5 wppm and 1 wppm depending on the microstructure). It is also for this reason that we have fitted the experimental total hydrogen concentration with a theoretical fit describing the charging of the specimens. Due to the fluctuation in experiments, the average total hydrogen concentration in LC+B and LC+LTT was larger of more than 1 wppm for 3h of charging time compared to 6h of charging time without a proper explanation. In the revised manuscript, measurements that are the furthest from the average values for LC, LC+B, LC+LTT, and LC+B+LTT have been removed when the charging time is 3h and 6h (when the microstructure is saturated with hydrogen). In this revised Figure 3.b of the manuscript, the difference between 3h and 6h is around 0.5 wppm, within the range of the standard deviation of the hydrogen concentration.

Reviewer's comment: 2. On page 9, Figure 4 compares the crack details of LC+H and LC+B+LTT+H to illustrate that the addition of element B resists hydrogen embrittlement. However, the variables here are not unique (LTT and B). If one wants to prove the effect of the addition of element B alone, the variable should be kept unique, such as by comparing LC+H with LC+B+H, or by comparing LC+LTT+H with LC+B+LTT+H.

Response: It is a good point from the reviewer. Therefore, we conducted additional crack analysis on LC+B+H and LC+LTT+H. Fig. 4 and the 4th part of the supplementary information have been modified accordingly. A slight variation of the percentages of crack types is observed, mostly because we reduced the step size to 0.05 μm , as hinted by reviewer #2. The crack propagation is reduced by tempering, most probably due to carbon segregation at PAGBs, as reported in the literature (Okada *et al.*) but boron seems to be more efficient on this microstructure feature. Also, adding boron mostly increases the fraction of cracks at interfaces, which happens at larger residual ductility. In the fourth part of the supplementary information, we have added all the EBSD data used in this study, besides modifying the 4th part of the revised manuscript as follows on page 9, lines 193 to 202:

"Figs. 4.a-d are **examples of** secondary cracks induced by the presence of hydrogen in LC, **LC+B, LC+LTT,** and LC+B+LTT, respectively. The distribution of the crystallographic or microstructural nature of these cracks is summarized in Fig. 4.e. For both steels, the measurement has been performed on more than 120 μm of cracks with images provided in the fourth part of the supplementary information. While more than half of the cracks observed in LC+H are from prior austenite boundaries, this fraction **had dropped to 20 % for the material variants LC+LTT+H,** suggesting an effect of carbon on reducing the crack propagation at these microstructure features in the presence of hydrogen, in agreement with previous results from the literature (Okada *et al.*). When boron is added, the fraction becomes lower than 10% for boron-doped variants."

Reviewer's comment: 3. While the suppression of prior austenite grain boundary cracking in LC+B+LTT is evident (Fig. 4c), the specific roles of boron/carbon segregation and stress relaxation in this process remain unclear.

Response: The reviewer is correct regarding the initial version of the manuscript, which was too elusive on the specific roles of solute and stress relaxation towards the change of behaviour observed through the secondary cracks analysis. To separate the contribution of boron and tempering, we pushed the analysis on LC+B and LC+LTT+B in fig. 4. Additionally, the synchrotron X-ray diffraction measurements can separate the contribution of carbon segregation and stress relaxation. The reduction of the austenite is mostly from stress relaxation, while the slight peak shift observed from untempered and tempered steels is from carbon segregation. In the revised manuscript, we highlight these contributions in fig. 2 and add on page 6 (lines 125 to 131):

“Fig. 2.f represents the determined lattice parameter from SXR D of both phases for all microstructures. When both LC and LC+B are tempered at 160°C for 4 h, the lattice parameter of the martensite is reduced by 0.017%, while it is reduced by 0.087% for the austenite. This lattice parameter reduction implies an increase in the compressive stress in the austenite, while it implies a stress relaxation of the martensite. It is mainly related to carbon diffusion (segregating from the matrix into the phase and grain boundaries) and partition during tempering. More information is available in the second part of the supplementary information on that topic.”

Additionally, we describe the specific role of boron and carbon segregation on pages 9 and 10, lines 202 to 210, as follows:

“Considering the *ab initio* calculations of fig. 1 and boron segregation previously measured through correlative TKD-APT measurements (in the second part of supplementary information), this fraction reduction underpins the tremendous improvement in HE resistance provided by boron. However, the weakest remaining microstructural features observed in materials LC+B+LTT+H are martensite boundaries and inclusions' interfaces, which fail at larger strains than the LC+H material variant. Therefore, the excess of carbon segregating at martensite boundaries only mitigates the embrittlement of these interfaces and does not completely stop hydrogen segregation at these interfaces. Further investigation has been performed to determine the type of inclusion.”

Finally, we performed ring core tests to estimate the relaxation stress when tempering is applied. This information is presented in the last part of the supplementary information.

Reviewer's comment: 4. The discussion on precipitates in the text is relatively simple. Please explain why the improvement of hydrogen embrittlement performance is not related to the hydrogen trap precipitates formed during tempering.

Response: We initially had a relatively short discussion in the first version of the manuscript because we previously reported the nucleation of carbides during the He gas quenching for LC and LC+B (Hachet *et al.*). These carbides were observed using ECC imaging and synchrotron measurements. By adding a tempering at 160°C to LC and LC+B, the same signal related to carbides is observed by SXR D, indicating a limited growth and/or nucleation of new carbides in the microstructure. Since transition carbides could increase the hardness of the microstructure (Song *et al.*, Takahashi *et al.*), we also performed hardness measurements, showing an increase of less than 1%, lower than the standard deviation of

the measurements (394 ± 10 HV₅₀₀ for LC+B and 397 ± 10 HV₅₀₀ LC+B+LTT). Consequently, the tempering does not form additional transition carbides due to the low amount of carbon (0.15wt.%) and the nucleation of carbides during auto tempering. From this comment and the 11th comment of the reviewer #3, we chose to add a paragraph related to this topic in the revised manuscript between 132 and 145, which is:

“Additionally, it has been reported that tempering at 160°C can induce the formation of transition carbides in both high-carbon and low-carbon steels (Cheng *et al.*, Kawahara *et al.* (2023)). These precipitates are known to trap hydrogen besides dislocations and grain boundaries and can significantly influence hydrogen embrittlement resistance [38, 39]. Therefore, the inset of Fig. 2.e highlights the contribution related to carbides (fluctuations of the diffraction pattern around 4.5°). The fluctuation is observed for all microstructures, indicating the presence of carbides for all our investigated samples. They are formed during the He quenching, after the martensite transformation (the so-called auto-tempering), and have been described previously for LC and LC+B (Hachet *et al.*). Further, micro-hardness measurements were performed to verify the formation of carbides (Cheng *et al.*, Kawahara *et al.* (2022)). The hardness of LC+B and LC+B+LTT were determined to be 394 ± 10 HV500 and 397 ± 10 HV500, respectively. The very similar hardness values suggest that the growth of these carbides or the formation of new transition carbides during the tempering is limited. Consequently, the trapping of hydrogen associated with these carbides should be similar with and without LTT. Additional information related to this topic is also provided in the second part of the supplementary information.”

In addition, in the second part of the supplementary information, we have added a zoom of the synchrotron measurements when $2\theta = \sim 4.5^\circ$ and $2\theta = \sim 7.5^\circ$ to show the similar behaviour of the signal related to carbides and obtained during the auto-tempering of the microstructure.

Hachet *et al.* : doi : 10.1016/j.ijhydene.2024.11.166

Cheng *et al.* : doi : 10.1007/bf02645469

Kawahara *et al.* (2023) doi : 10.1016/j.actamat.2023.118919

Takahashi *et al.*, doi : 10.1016/j.matchar.2021.111282

Huang *et al.* : doi : 10.1016/j.actamat.2025.121231

Kawahara *et al.* (2022) : 10.1016/j.matchar.2021.111579

Reviewer's comment: 5. The content of elements in martensitic and austenite is given in Figure 5c. Theoretically, carbon partitioning from martensite to austenite is expected to occur during low-temperature tempering. However, in Figure 5c, there is almost no difference in the C content between the austenite and martensite phases. Please check the accuracy of the data or provide an explanation.

Response: It is an important remark from the reviewer (that we should have discussed in the original version of the manuscript). The retained austenite is next to a grain boundary seen within the APT tip. Because our focus is primarily on the retained austenite and the manganese concentration, we initially chose not to present it. It is clearly a mistake, as it explains why there is no carbon partitioning in the retained austenite. In the revised manuscript, we added on page 11 (lines 239-248).

“One possible explanation is the higher concentration of carbon (close to 0.8 at%.) observed in the martensite and the $\gamma_{\text{Mn-rich}}$ phase, as the steel has been tempered at 160°C while it is closer to 0.2 at. %C. in the matrix and the $\gamma_{\text{Mn-lean}}$ phase. Additionally, one can note that a lower carbon concentration is observed after tempering in $\gamma_{\text{Mn-lean}}$ phase, even though carbon partitioning should have occurred. This result is obtained because of the carbon segregation at the grain boundary next to the γ phase, which is quantified in the fifth part of the supplementary information. Assuming a repulsive interaction between carbon and hydrogen

(which can be observed through *ab initio* calculations for GB in α -iron and seen at prior austenite grain boundary [43]), carbon could slow down hydrogen diffusion into this phase by increasing its concentration at the interface.”

Reviewer’s comment: 6. Figures 5e and 5f compare the transformation behavior of retained austenite with and without tempering before and after hydrogen charging, leading to the conclusion that "suggesting that hydrogen ingress is lower at martensite/austenite interfaces for tempered steels." However, this observation could also be explained by the increased stability of retained austenite during low-temperature tempering (a well-documented phenomenon in the literature), which consequently makes it more resistant to transformation under hydrogen charging. It is recommended that the authors provide a more comprehensive discussion regarding the behavior of retained austenite.

Response: It is a good argument from the reviewer, which needs to be discussed in the revised manuscript. In the updated version, we estimated the stress relaxation for the small peak shift noted by synchrotron X-ray diffraction. Because the shifts were small, we also performed complementary ring core experiments to estimate the internal stress released when the microstructure was changed before and after tempering. The details of this analysis are given in the fifth part of the supplementary information. For both approaches, a global hydrostatic stress relaxation of around -80 MPa is obtained. It results in a compressive stress of 76 MPa for the martensite and 374 MPa in the austenite. Therefore, the absence of early deformation-induced martensite transformation is also due to this effect. We thank the reviewer for highlighting this contribution.

In the revised manuscript, we have determined the resulting solubility reduction ($\Delta S_H^{\alpha',\gamma}$). It aims to qualitatively quantify the contribution of solute segregation and stress relaxation towards the reduction of the hydrogen ingress observed experimentally using thermal desorption spectroscopy. A reduction of ~10% of the hydrogen concentration should be seen when hydrogen is charged in tempered steel, but we observed a reduction of almost 50% experimentally. The difference can be explained by the effect of boron and carbon segregations at microstructure features, preventing hydrogen segregation. In the revised manuscript, we wrote between lines 259 and 284 (on pages 12 and-13):

“However, this change in austenite stability due to hydrogen is not observed when the microstructure is relaxed after tempering (see LC+LTT and LC+B+LTT in Fig. 5.f). It suggests that either hydrogen ingress or trapping is lower **within austenite** of tempered steels **or/and the austenite is becoming more stable after tempering.**

From Fig. 2, the tempering induces a slight reduction of the lattice parameter (Δ_{LTT}) for both martensite

Since no early deformation-induced martensite transformation is seen for hydrogen-precharged tempered steel, the retained austenite is either more stable (due to stress relaxation) or/the hydrogen uptake is reduced in this phase (possibly due to carbon segregation). To clarify this point, we determined the equivalent hydrostatic stress ($\sigma_{hydro}^{\alpha',\gamma}$) from the lattice parameter reduction ($\Delta_{LTT}^{\alpha',\gamma}$) using (Gong *et al.*):

$$\sigma_{hydro}^{\alpha',\gamma} = \frac{E^{\alpha',\gamma}}{1 - 2\nu} \Delta_{LTT}^{\alpha',\gamma}$$

With $E^{\alpha',\gamma}$ the isotropic Young's modulus of martensite and austenite ($E^{\alpha'} = 180$ GPa and $E^{\gamma} = 172$ GPa) and ν the Poisson's ratio ($\nu = 0.3$) (Gong *et al.*). Fig. 2 shows that the tempering induces a slight reduction of the lattice parameter for both martensite ($\Delta_{LTT}^{\alpha'} = -0.017\%$) and austenite ($\Delta_{LTT}^{\gamma} = -0.087\%$) from carbon partitioning (loss of carbon in solid solution from the

martensite) and a reduction of the austenite (due to stress relaxation). It leads to a relaxation of -76 MPa in the martensite ($\sigma_{hydro}^{\alpha'}$), and an increase of 374 MPa of the compressive stress in the austenite ($\sigma_{hydro}^{\gamma} = -374$ MPa). In the fifth part of the supplementary information, we have conducted ring core experiments on LC and LC+LTT, measuring a similar total hydrostatic stress. It strongly suggests that the tempering induces stress relaxation, as discussed in previous work (Zhao *et al.*), stabilizing the austenite. The resulting reduction of hydrogen ingress in austenite and martensite $\Delta S_H^{\alpha',\gamma}$ can be theoretically estimated using (Traisnel *et al.*):

$$\Delta S_H^{\alpha',\gamma} = \exp\left(\frac{\sigma_{hydro}^{\alpha',\gamma} \bar{V}_H^{\alpha',\gamma}}{k_B T}\right)$$

With $\bar{V}_H^{\alpha',\gamma}$ the partial volume of hydrogen in martensite and austenite ($\bar{V}_H^{\alpha'} = 3.3 \text{ \AA}^3/\text{at.}$ (Hirth) and $\bar{V}_H^{\gamma} = 2.0 \text{ \AA}^3/\text{at.}$ (Moody *et al.*)), k_B the Boltzmann constant and T the temperature. The resulting solubility reduction in martensite and austenite are $\Delta S_H^{\alpha'}$ is 94 % and ΔS_H^{γ} is 83 %. However, Fig. 3.b shows that the hydrogen concentration of the tempered LC is 60% for LC and 50 % for LC+B, which can be explained by the reduction of hydrogen ingress from boron and carbon segregation at interfaces.”

Gong *et al.*: doi: 10.1016/j.actamat.2023.118860

Zhao *et al.*: doi: 10.1016/j.scriptamat.2025.116789

Traisnel *et al.*: doi: 10.1016/j.commsci.2020.110136

Hirth: doi: 10.1007/BF02654700

Moody *et al.*: doi: 10.1016/S0036-9748(88)80143-1

Reviewer’s comment: 7. Given that existing literature has well documented the beneficial effects of certain elements (e.g., C) segregating at interfaces—enhancing interfacial cohesion energy and inhibiting the decohesion (regardless of hydrogen charging)—the authors should more prominently highlight the novel aspects of their work in this context.

Response: As suggested by the reviewer, we highlight the novel aspect of our work at the beginning and the end of the article. In the revised manuscript, the following sentence is added on page 3 of the revised manuscript between lines 38 and 43:

“Additionally, they can also probe the cohesive energy of solute-doped GB (E_{X-GB}^{coh}) relative to the undoped interfaces [25, 26]. Although numerous works have already been developed to highlight the effect of solutes to strengthen grain boundaries, information on their effect towards hydrogen segregation remains scarce, and experimental evidence of this behavior is lacking in the literature, which is the aim of this study.”

In addition, this sentence is added in the last paragraph of the manuscript:

“To summarize, atomistic calculations suggested that interstitial solutes, particularly boron and carbon at grain boundaries, have a tremendous effect against hydrogen segregation due to their capacity to strengthen interfaces and a repulsive interaction concerning hydrogen. While solute strengthening of interfaces in steel has been well documented in the literature, the influence of solute strengthening on hydrogen segregation has been less explored (some examples have been detailed in the design treasure maps of Fig. 1.c.). In the present work,

we successfully demonstrate experimentally how these insights can be used to design martensitic steels,..."

Reviewer's comment: 8. There are many writing flaws that make this article less readable : When an abbreviation appears for the first time in the main text, the complete term should be given, such as "GB" in the third paragraph; Each symbol in the formula should be given an explanation, such as NX in Formula (1); In the comments below Formula (1), two ESC X-FS are repeated, etc.

Response: We thank the reviewer for the careful reading of the manuscript and advise correcting the flaws. Corrections have been made in the manuscript and the supplementary information.

Reviewer #2

Reviewer's comment: 1. In Fig. 1c, the energy spectrum of B data exhibits a wide spread, similar to several other elements in the 'strengthening' scheme. Besides the distance between B and the solute atom (discussed on page 3), could the type of GB also be a factor? The authors have listed GB information in the Supplementary Data. Would it be helpful to incorporate GB type into the plot?

Response: The reviewer is right, the difference in the interaction energy between hydrogen and a boron-doped GB is quite large. In our work, we noted a difference of 0.36 eV when hydrogen is inserted at different positions around boron in the same GB. In the work of Song *et al.*, they noted a pronounced difference of almost 1 eV. It is how we conclude that the difference is mostly coming from the distance between the hydrogen and the doped element. However, the GB type can also affect the interaction energy as it already has an impact on the cohesive energy between the doping element and the GB. Therefore, in the revised manuscript, we add the type of GB in Fig. 1.c from different works in the captions and add in lines 62 and 66 (page 4):

"Additionally, ab initio calculations have also been performed in the present work for a range of solutes (boron, carbon, and nitrogen) in a $\Sigma 5(210)$ α -Fe GB with and without hydrogen.

One can note that the energy E_{H-X-GB}^{inter} is sensitive to the distance between hydrogen and the doping solute X within the same GB [17, 18], but also by different types of GB. Consequently, it leads to an energy spectrum for one doping solute, as presented in Fig. 1.c."

In the previous version, we also forgot to add the interaction energy between hydrogen and a nitrogen-doped GB and the cohesive energy of nitrogen with a GB from the work of Kholobina *et al.*, which is added in Fig. 3.c of the revised manuscript. This information becomes relevant because with nitrogen, our calculations show a slight strengthening of a $\Sigma 5$ while the work of Kholobina *et al.* and Matsumoto *et al.* shows an embrittlement of a $\Sigma 3$.

Song *et al.*: doi 10.1016/j.apsusc.2024.159684

Kholobina *et al.* doi: 10.1016/j.commat.2020.110215

Matsumoto *et al.* doi : 10.15669/pnst.2.9

Reviewer's comment: 2. Fig. 3b shows that the saturated hydrogen concentration nearly halves after LLT. Could the authors clarify this observation?

Changes in retained austenite or dislocation density seem insufficient to explain such a large difference. Will it be possible to further analyse the TDS data and draw useful conclusions from the perspective of quantifying trapping energies, and connect H content with microstructural feature.

Response: The reviewer is correct regarding the reduction of the hydrogen solubility after tempering LC and LC+B, which cannot be explained by the changes in retained austenite and dislocation densities. Regarding the dislocation density of the austenite, we noted that the peak related to (111) γ was not well indexed in the previous version of the manuscript, and was wrongly modifying the dislocation density in austenite. After correction, the dislocation density is similar for all conditions (untempered and tempered) and is $3.8 \times 10^{15} \text{ m}^{-2}$. In 8th paragraph of the revised manuscript, we wrote accordingly between lines 112 and 123 (page 5):

“Such effects are also observed during tempering: Fig. 2.f shows a similar distribution of grain boundary types, with less than 15% of prior austenite grain boundaries for all conditions. Using the circular integration of the synchrotron experiments, we determined a similar dislocation density in martensite and austenite of $\rho^{\alpha'} = 1.2 \times 10^{15} \text{ m}^{-2}$ and $\rho^{\gamma} = 3.8 \times 10^{15} \text{ m}^{-2}$ for all microstructures (with or without boron and tempering). The dislocation density for the different systems has been determined with the Williamson-Hall approach [28, 32, 33] and indicates that recovery or recrystallisation did not rejuvenate the microstructure in both phases when the tempering temperature is 160°C. However, a reduction of 30% (2.2 vol.% to 1.5 vol.%) of the retained austenite is noted. This metastable phase should be localized at martensite lath boundaries, according to previous work in medium carbon steel [34]. The loss in austenite volume fraction can indicate a stress relaxation of the martensite, which has been observed in recent work for press-hardened steels (Zhao *et al.*).”

Consequently, the reduction of hydrogen solubility is mostly due to 2 factors: solute segregation impeding hydrogen from segregating in crystalline defects sensitive to hydrogen embrittlement (interface), and stress relaxation of both phases, which is estimated through ring core tests presented in the fifth part of the supplementary information.

To further analyze the TDS data, temperature ramping is needed to determine the hydrogen desorption rate and not melting, as we performed here. We initially chose the melt extraction method because it gives more accurate results on the total hydrogen concentration (contrary to temperature ramping), but cannot give any information about the trapping energy or distribution of hydrogen in different traps. We have recently performed temperature ramping on water-quenched and tempered steels (which do not exactly have the same microstructures as the ones presented in this work). While it is still an ongoing work, it gives consistent results: the solubility of hydrogen is reduced with the tempering.

Figure: Hydrogen desorption rate of water-quenched low-carbon steel (untempered and tempered).

One can note that this result is slightly different from the recent work of Zhao *et al.* and more investigation is needed before publication of this result.

Zhao *et al.*: doi 10.1016/j.scriptamat.2025.116789

Reviewer's comment: 3. I assume the EBSD was performed over a large area to improve the reliability of fracture statistics. However, the quality of the maps in Fig. 4 could be improved. Reducing the pixel size would enable more detailed analysis.

Response: It is an excellent suggestion from the reviewer that we followed. In the revised manuscript, the analysis has been done by reducing the step size to 0.05 μm . Additionally, we have extended the analysis to LC+LTT+H and LC+B+H, as hinted by reviewer #1, to have a more complete study. The complementary information has been given in the method part of the manuscript as follows:

“The correct detection of the austenite was proven by the presence of a Kurdjumov-Sachs orientation relationship between the determined austenite and the neighboring martensite. Then, the crack analysis has been conducted on EBSD maps acquired using a Helios 5 FFIB (ThermoFischer) equipped with a CMOS detector. An acceleration voltage of 15 kV, probe current of 6.2 nA, and step size of 0.05 μm have been chosen for the mappings. Then, the MTEX 6.2.beta.3 software toolbox based on MATLAB R2023b was used to identify the type of secondary cracks on more than 120 μm of crack for each sample.”

Reviewer's comment: In the APT dataset, the shape of the austenite phase follows the curvature of the reconstructed 3D model. This raises questions about the reconstruction approach. In addition to the introduction of hydrogen during sample preparation and residual hydrogen, 1Da signals can also originate from environmental hydrogen in the APT chamber. Given the low content and surface-biased distribution, it is difficult to extract meaningful insights about hydrogen trapping. The authors are advised to interpret the hydrogen signal with more caution.

Response: The reviewer is correct that the shape of the austenite phase follows the curvature of the reconstructed 3D model, and as presented in the original version of the manuscript, it can raise questions about the interpretation of our results. This curvature is observed because we used the reconstruction from Geiser *et al.* protocol (Gault *et al.*), implemented in AP Suite. This protocol is known to develop distortions at the border of the tip, which is seen in Fig 5.b. However, this protocol does not change the chemical composition measurement; it only changes its size. In the revised manuscript, we have added complementary information on page 11 (lines 220-225), as follows:

“Fig. 5.b plots the reconstruction of an APT analysis of this thin film of austenite. This austenite follows the curvature of the 3D reconstruction, due to the reconstruction protocol from Geiser *et al.* implemented in the AP Suite software and known to develop distortion at the border of the reconstruction (Gault *et al.*). While this approach can modify the size of the thin film, it will not change the chemical composition plotted in Fig 5.c, which shows two ranges of manganese concentration.”

Additionally, we agree with the reviewer on difficulties in extracting meaningful insights about hydrogen trapping without a proper deuterium charging and cryo-transfer, which has not been done for this reconstruction. It is exactly for this reason that the information given is only qualitative and not quantitative. To separate residual hydrogen introduced before the APT analysis and hydrogen from residual gas, we estimate the electrostatic field by representing the relative ratio between the different charge states of iron ($Fe^+/(Fe^+ + Fe^{2+})$). The decrease of this relative ratio indicates a higher field, which should result in less hydrogen from ionized residual gas, and the opposite is observed in Fig. 5.c. In the revised manuscript, we extended our argumentation on page 11; lines 228-235:

“

The latter solute was not pre-charged; thus, it originates from either the sample preparation or it has already been present in this phase [49, 50] and can only give qualitative information on the hydrogen distribution. In the fifth part of the supplementary information, the evolution of the electrostatic field along the 1D concentration profile is presented using the relative charge state ratio of $Fe^+/(Fe^+ + Fe^{2+})$. The decrease of this ratio indicates an increase in the electrostatic field in the austenite, which should result in a reduction of hydrogen from residual gas ionization. Because a higher hydrogen concentration is detected in the austenite, the measured hydrogen concentration does not originate from this ionization.”

Gault *et al.*: doi: 10.1016/j.ultramic.2010.11.016

Chang *et al.* : doi : 10.1016/j.actamat.2018.02.064

Reviewer's comment: In the caption of Fig. 5, the phrase “local strain before deformation” may be a typo and should perhaps read “local deformation before fracture.”

Response: It is indeed a typo that has been modified in the revised version of the manuscript; we are thankful to the careful checking of the reviewer.

Reviewer's comment: In the Supplementary Data, should Tables 6 and 7 appear back-to-back? They are currently separated by Part 2.

Response: It is an accurate comment from the reviewer that could hinder the reading of the supplementary information, and it has been modified to have them back-to-back.

Reviewer's comment: In Fig. 5e, the in-situ synchrotron results show much more pronounced martensitic transformation in the H-charged, post-LTT sample. Could the authors elaborate on this observation?

Response: In this work, all the synchrotron experiments were performed *ex-situ*. In the revised manuscript, we made sure that we are not employing the term *in situ* to discuss our synchrotron measurements in the revised manuscript.

The comment is pertinent as we did not discuss this observation in the manuscript. When strained, we observed that a fraction of austenite is higher for LC+LTT+H (1.4%) than for LC+LTT (0.9%). There should not be any physical reason to observe more metastable austenite in hydrogen-charged and strained martensitic steel. We think it is mainly due to the accuracy of the measurements.

Reviewer #3

Reviewer's comment: 1) Please ensure that the full names of all abbreviations (e.g., HEDE, HELP, and so on) are clearly stated upon first use in the manuscript.

Response: The full name of the most used mechanism to describe hydrogen embrittlement is stated in the first page of the revised manuscript between lines 6 and 9, as follows:

“Mechanisms have been proposed to explain and predict **hydrogen embrittlement**, including **hydrogen enhanced decohesion (HEDE)**, **hydrogen enhanced localized plasticity (HELP)**, **absorption-induced dislocation-emission (AIDE)**, **hydrogen enhanced strain-induced vacancies (HESIV)**, **hydrogen induced phase transformation (HIPT)**, and so on.”

Reviewer's comment: 2) A more accessible explanation of the Interaction energy (E_{X-GB}^{inter}) and cohesive energy (E_{X-GB}^{coh}) would be helpful for readers who are not familiar with these concepts. While a detailed description appears to be included in the first part of the Supplementary information, a brief explanation should be provided in the main text, along with a clear reference directing readers to Supplementary Section 1 for further details.

Response: Since our manuscript aims to be read by different scientists, we really appreciate this remark. Additional information has been provided in the revised manuscript to help readers not familiar with these concepts. On page 4, between lines 51 and 61 of the revised manuscript, we have also added more details to improve the readability of fig. 1.c:

“Fig. 1.c plots E_{X-GB}^{coh} , which represents the cohesive energy of a GB with different solutes X as a function of E_{H-X-GB}^{inter} , which corresponds to the interaction of hydrogen with a GB doped with a solute, for different coherent site lattice of α -Fe GB [17-22, 27, 28]. Complementary *ab initio* calculations for B, C, and N on $\Sigma 5(210)$ α -Fe GB have been performed in this work, highlighting four specific regions depending on E_{X-GB}^{coh} and E_{H-X-GB}^{inter} . The first region is when $E_{X-GB}^{coh} < 0$ and $E_{H-X-GB}^{inter} > 0$: it is where the best solutes against HE should be located since a strengthening of the GB with a repulsive interaction against hydrogen is observed by doping GB with X. The second one is when $E_{X-GB}^{coh} < 0$ and $E_{H-X-GB}^{inter} < 0$, where the solute X has a limited impact on HE because it strengthens GB, but hydrogen is still attracted to GB. The third one is when $E_{X-GB}^{coh} > 0$ and $E_{H-X-GB}^{inter} > 0$, where the solute X is inefficient because it has a repelling effect against hydrogen, but it embrittles GB. Finally, the last region is when $E_{X-GB}^{coh} > 0$ and $E_{H-X-GB}^{inter} < 0$: it is where the solute X is detrimental due to an embrittlement of the interface in combination with an attraction of hydrogen to the GB. More information related

to this figure, including the different types of GB, is provided in the first part of the supplementary information.”

Reviewer’s comment: 3) In the explanation of Fig. 1c, the authors state: “The GB energy design map reveals that boron, carbon, molybdenum, tungsten, niobium, and chromium are the best elements because they increase the cohesion of GB and can have a repulsive interaction with hydrogen.” However, several of these elements appear to fall at least partially within the attraction region on the map. It would be helpful if the authors could provide additional clarification on the criteria used to make this assessment.

Response: The comment from the reviewer is pertinent: with the simple explanation provided in the manuscript, aluminium, silicon, titanium, and manganese are also within the criteria. It is also the case for nitrogen if we are omitting the results on Σ_3 GB. Initially, we focused our comparison more on the cohesive energy than the interaction energy, because of the large discrepancy between the interaction energies. Therefore, we have selected the six most suited elements, which can have a repulsive interaction energy with hydrogen and strengthen the crystalline defect. However, this comparison does not rigorously describe the results from Fig 1.c. Consequently, we decided to write a paragraph on it between lines 64 and 78 of pages 4 and 5 as follows:

“For the approach developed in this study, the most suited solute against hydrogen embrittlement should increase the cohesion of the GB and have a repulsive interaction with hydrogen when inserted into the crystalline defect to repel hydrogen. Guided by these metrics, boron becomes the best element because it can have a repulsive energy (E_{H-X-GB}^{inter}) above 0.5 eV and a cohesive energy (E_{X-GB}^{coh}) around -1 eV. These high hydrogen repulsion and cohesion values would provide the GB with mechanical strengthening and chemical protection against hydrogen. Niobium and carbon are also potentially good candidates due to a repulsive interaction energy between 0.2 eV and 0.5 eV at a cohesive energy around -0.5 eV. Molybdenum and tungsten would also be suited elements specifically for strengthening the GB (E_{X-GB}^{coh} below -0.5 eV), but their interaction energies are at maximum close to 0 eV. Finally, aluminum, silicon, titanium, chromium, and manganese also fall within the criteria of suited protective elements against hydrogen but have a cohesive energy close to 0 eV with an interaction energy which can be at maximum at 0.2 eV.”

Reviewer’s comment: 4) Figures 1d-e would benefit from a more reader-friendly explanation. For instance, it could be helpful to explicitly state that a more negative mixing enthalpy implies a stronger tendency for bonding. Additionally, the statement that “carbon and nitrogen are the most reactive elements with iron” appears somewhat ambiguous. Phosphorus, which is also an interstitial element, shows a similar trend to carbon, making the distinction unclear. Furthermore, the claim that “tungsten and niobium are even most reactive than iron to form carbide” seems inconsistent with the behavior of other elements such as Zr, Re, Ta, and Ti, which also appear highly reactive based on the data, yet are not discussed. Finally, the concluding remark that “boron is the most suited solute against hydrogen segregation at GB, in addition to molybdenum and carbon” feels somewhat abrupt and unsubstantiated, given the preceding analysis. A more coherent and supported interpretation is recommended.

Response: It is a fair point from the reviewer. In the revised manuscript, we focus our comparison only on solutes that are good candidates against hydrogen segregation when they are introduced as solute in the GB, which is why the investigation was only focused on B, C, Cr, Nb, Mo, and W. By being more rigorous in the selection of the most suited solutes against hydrogen segregation in the revised manuscript, we extended the study to Al, Si, Ti,

and Mn which could have a slightly good repelling effect toward hydrogen segregation at GBs. Consequently, in lines 82 to 98 of the manuscript is written as follows:

“Therefore, Figs. 1.d,e present additional design maps plotting the mixing enthalpy of X with iron and carbon ($H_{X-(Fe,C)}^{mix}$) as a function of the interaction between X and a GB (E_{X-GB}^{inter}). A positive $H_{X-(Fe,C)}^{mix}$ indicates a preferential phase separation between the solute X and iron or carbon, whereas a negative $H_{X-(Fe,C)}^{mix}$ indicates a tendency to form a precipitate. We also chose to investigate the mixing enthalpy of X with carbon because it is present at a high concentration in a GB for steels. Figs. 1.d,e shows that all solutes are attracted to the interface (because all E_{X-GB}^{inter} are negatives). While solutes like boron, phosphorus, and silicon have a low mixing enthalpy ($H_{B-Fe}^{mix} = -0.37$ eV, $H_{P-Fe}^{mix} = -0.41$ eV, and $H_{Si-Fe}^{mix} = -0.36$ eV), carbon and nitrogen are the most reactive elements ($H_{C-Fe}^{mix} = -0.52$ eV and $H_{N-Fe}^{mix} = -0.90$ eV). This finding suggests that no precipitates will be formed if iron carbide and nitride formation are avoided. However, Fig. 1.e shows that with carbon, niobium and tungsten are more reactive than iron to form carbides ($H_{Nb-C}^{mix} = -0.56$ eV and $H_{W-C}^{mix} = -0.62$ eV). Consequently, boron seems to be the most suited solute against hydrogen segregation at GB in addition to carbon. Considering the strengthening effect of each solute element, molybdenum is also a good candidate, even though the repulsive energy can be at a maximum close to zero, and solutes like aluminium, silicon, chromium, and manganese should have a slightly positive effect against hydrogen segregation.”

Reviewer’s comment: 5) In the initial description of the experimental procedures, the purpose of the low temperature tempering (LTT) treatment should be clearly articulated. Since readers are likely to focus on the role of boron and carbon from the beginning of the manuscript, the sudden introduction of LTT may seem abrupt or confusing. It is therefore recommended that the rationale behind the LTT treatment be explicitly stated at the point where it is first mentioned, to provide better context and improve the logical flow.

Response: We thank and agree with the reviewer on this point: giving the reasons for the LTT treatments when presenting the different microstructures is something that should be in the manuscript. Therefore, in lines 103 to 108 (page 5) of the revised manuscript, we wrote:

“Consequently, four different microstructures have been investigated and are referred to as LC, LC+B, LC+LTT, and LC+B+LTT (the +B implies that boron has been added, and +LTT implies that the steel has been subjected to low-temperature tempering at 160°C). While boron addition and segregation should mainly strengthen prior austenite grain boundaries, the LTT will have two impacts against HE. First, it induces carbon segregation at various interfaces. Second, it should reduce the internal stresses resulting from quenching-induced martensitic transformation.”

Reviewer’s comment: 6) In Fig. 2f, the method used to determine the distribution of grain boundary (GB) types should be described more precisely. Additionally, the authors should briefly explain the reason behind the observed decrease in austenite fraction after the LTT treatment. Moreover, it is important to clarify how the dislocation densities of martensite and austenite phases were individually measured or calculated for each specimen. If a detailed explanation is not possible within the main text, at the very least, the authors should provide a citation to a reference that employed the same or a similar methodology.

Response: The first version of the manuscript was indeed too quick in explaining some important details, such as the identification of grain boundaries or the dislocation density determination. In the revised manuscript, we have added how the separation of the three

types of grain boundaries in martensite is done in the “electron back scattered diffraction analysis” section of the methods part:

“Then, the MTEX 5.11.1 software toolbox based on MATLAB R2021b was used to identify the misorientation angle and axis of all grain boundaries to reconstruct prior austenite boundaries from the EBSD maps [60]. Grain boundaries were separated into three types: (i) Martensite boundaries that are grain boundary variants formed with the Kurdjumov-Sachs orientation relationship, (ii) Prior austenite grain boundaries that are others random boundaries with a misorientation angle larger than 8°, and (iii) sub-grain boundaries, which are low-angle grain boundaries, usually representing martensite lath boundaries [28]. Spherical indexing was used to analyze EBSD patterns that were recorded with a 10 nm step size at a low acceleration voltage of 10 kV.”

The determination of the dislocation density was individually measured on different heat-treated specimens. The dislocation density in each phase was then calculated from the position of each peak and its full-width at half maximum, in agreement with the Williamson-Hall approach. In the revised version, we added in the method section additional details:

“A high-energy transmission X-ray beam with a wavelength of 0.207381 Å was shed on square-shaped specimens (10x10x1 mm³) to collect two-dimensional diffractograms at a working distance of 969 mm. All specimens were individually heat-treated in Bähr DIL805 dilatometer, and aged for one month at room temperature before performing the SXRD measurements on each specimen. Before conducting quantitative diffraction analyses, the instrumental parameters have been calibrated using the diffraction patterns of NIST standard LaB₆. All recorded two-dimensional diffractograms have been post-processed using the GSAS-II software [52]. The fraction of austenite has been determined by integrating peaks from both austenite and martensite, and the dislocation density ρ in both phases has been determined using the Williamson-Hall approach [35] on the different tempered specimens:”

Reviewer’s comment: 7) The sentence “This effect is mostly due to the presence of boron in prior austenite boundaries” would benefit from a clear reference to Supplementary Fig. 2, as it appears to provide supporting evidence for this claim. Explicitly directing the reader to the relevant figure will help strengthen the author’s argument and improve clarity.

Response: We agree with the reviewer on this comment. In the revised manuscript, we add on page 8 (lines 163 and 166):

“This strongly suggests an improved resistance against HE, mostly due to the boron segregation at prior austenite grain boundaries. This improvement has been characterized previously using correlative TKD-APT measurements (Hachet *et al.*, Shi *et al.*) and shown in the second part of the supplementary information.”

Hachet *et al.*: doi : 10.1016/j.ijhydene.2024.11.166

Shi *et al.*: 10.1016/j.ijhydene.2025.150954

Reviewer’s comment: 8) In Fig. 3c, the point-curve graph presents data with a relatively wide range indicated around the curves. It is important to clarify what this range represents – whether it reflects experimental variability, standard deviation, confidence intervals, or some other metric. Providing this explanation will help readers interpret the data more accurately.

Response: The reviewer is correct about the relatively wide range observed in fig 3.b. To better interpret the data, we have updated the figure in the revised manuscript by showing all

measurements. Additionally, while the accuracy of the thermal conductivity detector is lower than 0.1 wppm, fluctuation up to 1 wppm can be observed due to the charging condition. Consequently, the standard deviation determined is measured by considering the highest range to encompass all measured and fitted data. This information has been added in the method section of the revised manuscript as follows:

“The hydrogen concentration C_l is the limit fitted to the hydrogen concentration determined by thermal desorption spectroscopy. The standard deviation has been determined to encompass all measured and fitted concentrations per microstructure, resulting in a standard deviation of 0.9 wppm for LC, 0.7 wppm for LC+B and LC+LTT, and 0.6 wppm for LC+B+LTT.”

Reviewer’s comment: 9) The statement “Previous work has shown that a boron addition impedes crack propagation along prior austenite boundaries, as depicted in Fig. 1.b [9, 12].” Would benefit from a brief explanation of the underlying mechanism. Even a concise summary of how boron contributes to impeding crack propagation – such as through grain boundary segregation, strengthening, or chemical interaction – would help readers better understand the context and significance of this finding.

Response: We agree with the reviewer on this comment. On pages 8 and 9 of the revised manuscript, we wrote in lines 188-191:

“Previous works have shown that a boron addition impedes crack propagation along prior austenite grain boundaries [13,16], due to an increase of grain boundary cohesion by a strong hybridization between the iron s, d-states, and the boron s, p-states, according to previous *ab initio* calculations [21,42]. Further, we opted...”

Reviewer’s comment: 10) Despite identical electrocharging conditions before and after LTT, there appears to be a substantial difference in hydrogen ingress. While it is reasonable to expect that hydrogen embrittlement sensitivity can vary due to changes in hydrogen trapping sites, it is more difficult to understand why the total amount of hydrogen ingress itself would differ significantly under the same charging conditions for similar steel-based specimens. This discrepancy requires clear explanation. Including permeation test results would substantially strengthen the author’s argument; if such data are unavailable, a more detailed rationale should still be provided to justify this observation.

Response: It is a fair point from the reviewer. The reduced hydrogen concentration is due to reduced solubility, partially due to stress relaxation, and also due to carbon segregation at crystalline defects such as martensite boundaries. In the revised manuscript, we have developed the discussion regarding this result in the last paragraph before summarizing the results developed in this work. We hoped it would give a clear explanation for the reader on the effect of the microstructure on the reduction of the hydrogen solubility. As suggested by the reviewer, we have also performed hydrogen permeation tests on boron-free steels, with and without tempering, which is presented in the third part of the supplementary information. The steady state permeation rate (i_{∞}) is slightly lower for the tempered steel (LC+LTT) compared to the untempered one (LC), which implies a reduced solubility, observed by TDS. We also noted that the diffusion of hydrogen is faster in LC+LTT compared to LC. Knowing that tempering induces stress relaxation and carbon partitioning, it is also consistent with our TDS experiments. Therefore, between lines 177 and 185 of pages 8 and 9 of the revised manuscript, we have added:

“Complementary experiments have also been conducted for more severe hydrogen charging conditions, still showing an improved resistance against HE of the LC+B+LTT compared to the other microstructure (third part of the supplementary information). Hydrogen permeation

tests (presented in the third part of the supplementary information) show a slightly reduced steady-state permeation rate, suggesting a reduced hydrogen solubility due to tempering also confirmed in Fig. 3.b. An increase in the apparent hydrogen diffusion coefficient is observed when the steel is tempered. It might be induced by the internal stress relaxation from carbon segregation and austenite reduction, reducing the number of trapped sites, which is also consistent with our TDS experiments.”

Reviewer’s comment: 11) The LTT condition employed in this study – 160 oC for 4 hours – is known to allow the formation of epsilon (ϵ) -carbide in martensitic structures. Since ϵ -carbide are generally regarded as relatively strong hydrogen trapping sites, they could significantly influence hydrogen embrittlement resistance. However, the manuscript does not provide any discussion or experimental analysis related to carbide precipitation before and after LTT. Given that strong hydrogen trapping sites such as ϵ -carbide can significantly influence hydrogen embrittlement resistance, it is essential not only to investigate the carbide evolution before and after LTT, but also to explicitly consider how such carbide formations may alter the overall resistance to hydrogen embrittlement. Experimental data or at least a discussion on this aspect is essential to support the interpretation of hydrogen-related behavior. Relevant references include: ① Transition from carbon clusters to ϵ , θ -carbides in a quenched and aged low-carbon ferritic steel, and ② Atomic-scale observation of hydrogen trap sites in bainite–austenite dual-phase steel by APT.

Response: It is a good point from the reviewer due to the lack of information related to carbide formation in the initial version of the manuscript. Previous reported work indeed shows that this LTT condition can allow the formation of transition carbides. However, these carbides are usually observed in steels containing a higher amount of carbon than the system investigated in our work (Cheng *et al.*), or water-quenched low-carbon steel (Takahashi *et al.*). He gas quenching favors precipitation during autotempering, which is less observed for water-quenched systems. Consequently, the driving force to nucleate new transition carbides is too low to observe other carbides, or at least see a visible effect of these carbides formed during tempering. Consequently, similar hardness and synchrotron measurements are observed for steel with and without tempering. From this comment and the 4th comment of reviewer #1, we chose to add a paragraph related to this topic between 132 and 145, which is:

“Additionally, it has been reported that tempering at 160°C can induce the formation of transition carbides in both high-carbon and low-carbon steels (Cheng *et al.*, Kawahara *et al.* (2023)). These precipitates are known to trap hydrogen besides dislocations and grain boundaries and can significantly influence hydrogen embrittlement resistance [38, 39]. Therefore, the inset of Fig. 2.e highlights the contribution related to carbides (fluctuations of the diffraction pattern around 4.5°). The fluctuation is observed for all microstructures, indicating the presence of carbides for all our investigated samples. They are formed during the He quenching, after the martensite transformation (the so-called auto-tempering), and have been described previously for LC and LC+B (Hachet *et al.*). Further, micro-hardness measurements were performed to verify the formation of carbides (Cheng *et al.*, Kawahara *et al.* (2022)). The hardness of LC+B and LC+B+LTT were determined to be 394 ±10 HV500 and 397±10 HV500, respectively. The very similar hardness values suggest that the growth of these carbides or the formation of new transition carbides during the tempering is limited. Consequently, the trapping of hydrogen associated with these carbides should be similar with and without LTT. Additional information related to this topic is also provided in the second part of the supplementary information.”

In addition, in the second part of the supplementary information, we have added a zoom of the synchrotron measurements when $2\theta = \sim 4.5^\circ$ and $2\theta = \sim 7.5^\circ$ to show the similar behaviour of the signal related to carbides and obtained during the auto-tempering of the system.

Hachet *et al.* : doi : [10.1016/j.ijhydene.2024.11.166](https://doi.org/10.1016/j.ijhydene.2024.11.166)
Cheng *et al.* : doi : [10.1007/bf02645469](https://doi.org/10.1007/bf02645469)
Kawahara *et al.* (2023) doi : [10.1016/j.actamat.2023.118919](https://doi.org/10.1016/j.actamat.2023.118919)
Takahashi *et al.*, doi : [10.1016/j.matchar.2021.111282](https://doi.org/10.1016/j.matchar.2021.111282)
Huang *et al.* : doi : [10.1016/j.actamat.2025.121231](https://doi.org/10.1016/j.actamat.2025.121231)
Kawahara *et al.* (2022) : [10.1016/j.matchar.2021.111579](https://doi.org/10.1016/j.matchar.2021.111579)

Reviewer's comment: 12) In Fig. 4c, the authors present results only for the LC+H and LC+B+LTT+H specimens. It would be helpful to explain why these two specific samples were selected for analysis. If data are available for all four specimen conditions, providing the distribution of SC classifications for the full set would offer a more comprehensive understanding and strengthen the overall argument.

Response: We initially thought the analysis would be easier to present by comparing the secondary crack of hydrogen-charged steels with and without boron and tempering, which was a mistake. In the revised manuscript, we present the secondary cracks of all specimen conditions. We highlight the effect of boron and tempering on prior austenite grain boundaries and the increase of cracks at interfaces when the plastic strain is larger for the boron-added and tempered steel.

Reviewer's comment: 13) It is recommended that MnS in Figs. 4d-f be explicitly labeled as an inclusion, as this terminology more accurately reflects its metallurgical nature.

Response: We have followed the recommendation of the reviewer in fig. 4.d-f and also update the revised manuscript to define MnS as inclusions

Reviewer's comment: 14) For Fig 5.c, it is recommended – if feasible – to include data showing changes in hydrogen trapping behavior after hydrogen charging. Such information would greatly enhance the credibility and depth of the analysis, providing stronger support for the author's interpretation.

Response: The reviewer is correct: such information would indeed provide stronger support for our work, but it is extremely challenging. It would require a site-specific preparation to isolate retained austenite, which should be verified using transmission Kikuchi diffraction (TKD) before charging the APT tip with deuterium (to separate hydrogen ionized from residual gas in the analysis chamber) and performing a cryo-transfer to the atom probe tomography apparatus. However, TKD introduces a defect where deuterium is segregating, as discussed in the work of Saksena. *et al*, which would influence the trapping of deuterium. Fig. 5.c is not aimed at quantifying; it only aims to give qualitative information on residual hydrogen. Consequently, in the revised manuscript, we highlight this point on page 10, lines 228-230, as follows:

“The latter solute was not pre-charged; thus, it originates from either the sample preparation or it has already been present in this phase [49, 50] and can only give qualitative information on the hydrogen distribution.”

Saksena *et al.*: doi: [10.1016/j.ijhydene.2023.09.057](https://doi.org/10.1016/j.ijhydene.2023.09.057)

Reviewer's comment: 15) In Figs. 5e-f, the data suggest that deformation-induced martensitic transformation in austenite is significantly suppressed after LTT, indicating that the austenite has been stabilized. It would be beneficial to include a brief explanation of the

possible mechanisms behind this stabilization, such as carbon enrichment, changes in dislocation density, or retained austenite morphology.

Response: Following this comment and comments 3 and 6 from reviewer #1, we have performed additional ring-core experiments and pushed the analysis of our synchrotron data. There is both stress relaxation and reduction of the hydrogen uptake in both phases, which can explain the suppression of the early deformation-induced martensitic transformation. We estimated the stress relaxation in the austenite and the martensite in the revised manuscript and detailed it in the fifth part of the supplementary information. From them, a resulting solubility reduction can be evaluated, which is not in the same range as the reduction of hydrogen seen experimentally. It highlights the effect of boron and carbon segregations in microstructure features, preventing hydrogen segregation. In the revised manuscript, we wrote between lines 259 and 284 (on pages 12 and-13):

“However, this change in austenite stability due to hydrogen is not observed when the microstructure is relaxed after tempering (see LC+LTT and LC+B+LTT in Fig. 5.f). It suggests that either hydrogen ingress or trapping is lower **within austenite** of tempered steels **or/and the austenite is becoming more stable after tempering.**

From Fig. 2, the tempering induces a slight reduction of the lattice parameter (Δ_{LTT}) for both martensite

Since no early deformation-induced martensite transformation is seen for hydrogen-precharged tempered steel, the retained austenite is either more stable (due to stress relaxation) or/the hydrogen uptake is reduced in this phase (possibly due to carbon segregation). To clarify this point, we determined the equivalent hydrostatic stress ($\sigma_{hydro}^{\alpha',\gamma}$)

from the lattice parameter reduction ($\Delta_{LTT}^{\alpha',\gamma}$) using (Gong *et al.*):

$$\sigma_{hydro}^{\alpha',\gamma} = \frac{E^{\alpha',\gamma}}{1 - 2\nu} \Delta_{LTT}^{\alpha',\gamma}$$

With $E^{\alpha',\gamma}$ the isotropic Young's modulus of martensite and austenite ($E^{\alpha'} = 180$ GPa and $E^{\gamma} = 172$ GPa) and ν the Poisson's ratio ($\nu = 0.3$) (Gong *et al.*). Fig. 2 shows that the tempering induces a slight reduction of the lattice parameter for both martensite ($\Delta_{LTT}^{\alpha'} = -0.017\%$) and austenite ($\Delta_{LTT}^{\gamma} = -0.087\%$) from carbon partitioning (loss of carbon in solid solution from the martensite) and a reduction of the austenite (due to stress relaxation). It leads to a relaxation of -76 MPa in the martensite ($\sigma_{hydro}^{\alpha'}$), and an increase of 374 MPa of the compressive stress in the austenite ($\sigma_{hydro}^{\gamma} = -374$ MPa). In the fifth part of the supplementary information, we have conducted ring core experiments on LC and LC+LTT, measuring a similar total hydrostatic stress. It strongly suggests that the tempering induces stress relaxation, as discussed in previous work (Zhao *et al.*), stabilizing the austenite. The resulting reduction of hydrogen ingress in austenite and martensite $\Delta S_H^{\alpha',\gamma}$ can be theoretically estimated using (Traisnel *et al.*):

$$\Delta S_H^{\alpha',\gamma} = \exp\left(\frac{\sigma_{hydro}^{\alpha',\gamma} \bar{V}_H^{\alpha',\gamma}}{k_B T}\right)$$

With $\bar{V}_H^{\alpha',\gamma}$ the partial volume of hydrogen in martensite and austenite ($\bar{V}_H^{\alpha'} = 3.3 \text{ \AA}^3/\text{at.}$ (Hirth) and $\bar{V}_H^{\gamma} = 2.0 \text{ \AA}^3/\text{at.}$ (Moody *et al.*)), k_B the Boltzmann constant and T the temperature. The resulting solubility reduction in martensite and austenite are $\Delta S_H^{\alpha'}$ is 94 % and ΔS_H^{γ} is 83 %.

However, Fig. 3.b shows that the hydrogen concentration of the tempered LC is 60% for LC and 50 % for LC+B, which can be explained by the reduction of hydrogen ingress from boron and carbon segregation at interfaces.”

Gong *et al.*: doi: 10.1016/j.actamat.2023.118860

Zhao *et al.*: doi: 10.1016/j.scriptamat.2025.116789

Traisnel *et al.*: doi : 10.1016/j.commat.2020.110136

Hirth: doi : 10.1007/BF02654700

Moody *et al.*: doi: 10.1016/S0036-9748(88)80143-1

Reviewer #4

Reviewer's comment: I co-reviewed this manuscript with one of the reviewers who provided the listed reports. This is part of the Nature Communications initiative to facilitate training in peer review and to provide appropriate recognition for Early Career Researchers who co-review manuscripts.

Response: We hope that reviewing this manuscript is a good experience for the early-career researcher of the reviewer.

We cordially thank the Editor and the Reviewers for the time and effort taken to evaluate our manuscript. Below, please kindly find detailed answers to all the comments, suggestions, and questions. Changes are highlighted in red in the revised manuscript.

Reviewer #1

Reviewer's comment: Thank you to the authors for their thorough and thoughtful responses. All concerns have been adequately addressed.

Response: All the authors sincerely thank the reviewer for the time spent on it. All comments received were on point.

Reviewer #2

Reviewer's comment: The authors did a good job addressing all my comments. The quality of the paper has been significantly improved.

Response: We agree with the reviewer that the manuscript has been improved by addressing the concerns and comments from the reviewer. We thank you again for that.

Reviewer #3

Reviewer's comment: The authors have responded to the reviewer's comments in a thorough and sincere manner. The explanations provided are reasonable and generally acceptable from a scientific standpoint. Based on the revisions and clarifications made, I find the manuscript to be acceptable for publication.

- Ph.D. Hyungkyun Park

Response: Dear Dr. Hyungkyun Park, thank you for your accurate and thorough review, which helped us improve the manuscript.

Reviewer #4

Reviewer's comment: I co-reviewed this manuscript with one of the reviewers who provided the listed reports. This is part of the Nature Communications initiative to facilitate training in peer review and to provide appropriate recognition for Early Career Researchers who co-review manuscripts.

Response: Again, we hope that reviewing this manuscript is a good experience for the early-career researcher of the reviewer.